# Application of high-throughput single-nucleus DNA sequencing in pancreatic cancer

Haochen Zhang [1,2], Elias-Ramzey Karnoub [2,3], Shigeaki Umeda [2,4], Ronan Chaligné [5], Ignas Masilionis[5], Caitlin A. McIntyre[6], Palash Sashittal[7], Akimasa Hayashi [2,4,8], Amanda Zucker [2,4,9], Katelyn Mullen [1,2], Jungeui Hong[3,4], Alvin Makohon-Moore[2,4,10] & Christine A. Iacobuzio-Donahue [2,3,4] ✉

Despite insights gained by bulk DNA sequencing of cancer it remains challenging to resolve the admixture of normal and tumor cells, and/or of distinct tumor subclones; high-throughput single-cell DNA sequencing circumvents these and brings cancer genomic studies to higher resolution. However, its application has been limited to liquid tumors or a small batch of solid tumors, mainly because of the lack of a scalable workflow to process solid tumor samples. Here we optimize a highly automated nuclei extraction workflow that achieves fast and reliable targeted single-nucleus DNA library preparation of 38 samples from 16 pancreatic ductal adenocarcinoma patients, with an average library yield per sample of 2867 single nuclei. We demonstrate that this workflow not only performs well using low cellularity or low tumor purity samples but reveals genomic evolution patterns of pancreatic ductal adenocarcinoma as well.

The field of single-cell genomics, since its advent about 10 years ago[1,2], has been striving to increase the throughput and resolution of cancer research. Single-cell DNA sequencing (scDNA-seq) offers many advantages over traditional "bulk" DNA-seq. Most importantly, it circumvents the issue of "mixed signals"[3], i.e. the admixture of normal and tumor cells, and/or of distinct tumor subclones. Solving the former allows for much higher sensitivity in calling rare genetic events, which opens opportunities to validate and discover cancer-related somatic mutations. Solving the latter allows for more confident identification of different clonal lineages within a single tumor, which could

inform understanding cancer evolution as well as targeted treatment decisions.

Bulk sequencing of pancreatic ductal adenocarcinoma (PDAC) is particularly problematic because of the high stromal content and low-tumor cellularity which further lowers variant calling sensitivity[4–6]. Present solutions include multiregional sampling to increase sensitivity for variants with low allele frequency[7,8] or laser-capture tissue microdissection to enrich for tumor content[9,10], but they are laborious and not amenable to high-throughput. With targeted single-cell sequencing, because each cell is partitioned and PCR amplified

[1]Gerstner Sloan Kettering Graduate School of Biomedical Sciences, Memorial Sloan Kettering Cancer Center, New York, NY, USA. [2]Human Oncology and Pathogenesis Program, Memorial Sloan Kettering Cancer Center, New York, NY, USA. [3]David M. Rubenstein Center for Pancreatic Cancer Research, Memorial Sloan Kettering Cancer Center, New York, NY, USA. [4]Department of Pathology and Laboratory Medicine, Memorial Sloan Kettering Cancer Center, New York, NY, USA. [5]Computational and Systems Biology Program, Sloan Kettering Institute, Memorial Sloan Kettering Cancer Center, New York, NY, USA. [6]Department of Surgery, Memorial Sloan Kettering, New York, NY, USA. [7]Department of Computer Science, Princeton University, Princeton, NJ, USA. [8]Present address: Department of Pathology, Kyorin University School of Medicine, Mitaka City, Tokyo, Japan. [9]Present address: School of Medicine, Oregon Health and Science University, Portland, OR, USA. [10]Present address: Center for Discovery and Innovation, Hackensack Meridian Health, Nutley, NJ, USA. ✉e-mail: iacobuzc@mskcc.org

individually, high-quality genomic data from a low percentage of tumor cells could potentially be extracted from the background noise, making it valuable for genomic studies of PDAC.

To date, several high-throughput single-cell partitioning systems have been developed, including microfluidic platforms, nanowells and microdroplets, which have resulted in several reliable single-cell DNA library preparation technologies[11]. Tapestri[12,13], as a microdroplet-based, targeted sequencing approach, allows for high cell throughput (up to 10,000 cells per sample) and high coverage depth (>80X) of genomic sites of interest, and is therefore suited for high-resolution studies of key genetic variants within diseases. However, its use so far has been limited to cell lines and liquid primary tumor samples, or a small batch of solid tumor tissues at a time[14–20]. While methods to quickly and effectively dissociate clean single nuclei suspensions from solid tumor tissues have been extensively tested for single-nucleus RNA-seq (snRNA-seq)[21], they have not been applied for single-nucleus DNA-seq (snDNA-seq) usage.

In this work, we optimize a snap-frozen tissue single nuclei extraction workflow that yields high throughput in generating the resulting snDNA libraries. Importantly, the workflow takes <30 min per sample with minimal manual labor, thus ideal for processing large batches of solid tumor samples. Coupling the snDNA data with bulk whole exome sequencing (WES) or whole-genome sequencing (WGS) data generated on the same samples, we are able to uncover single-cell clonal relationships among key driver mutations.

## Results

### Optimization of workflow to extract and store single nuclei from snap-frozen tissue

We recognized the need for a nuclei extraction workflow that has reduced hands-on operation, sample resuspension times, and total processing time, all of which hinder scalability and could potentially cause between-sample inconsistency in quality (clumping, debris) and final yields. Thus, we used an automated nuclei extraction machine[22] for homogenizing frozen tissues into single nuclei suspensions. The nuclei were then passed through a sucrose gradient to strip away debris before microdroplet encapsulation. The entire procedure takes ~30 min per sample and requires a single step of pelleting and resuspension (Fig. 1a; "Methods"). Although the resulting nuclei clumping percentage and nuclei concentration varied across samples of different starting sizes, cellularity and morphology, most primary pancreas tumor samples of volumes ≥ 8 mm³, regardless of collection method (resection vs autopsy), resulted in final nuclei suspensions with ≤10% clumping and >2000 nuclei/μl suspended in > 35 μl buffer as input for Tapestri (Fig. 1b). Exceptions were samples with particularly high fat/stroma content or random technical errors, which either limited input nuclei concentration or increased clumping rate as observed by microscopy.

With this protocol, we prepared 38 snDNA libraries from 34 biologically distinct snap-frozen PDAC samples from 16 patients. These samples were purposely selected to represent primary tumors and metastases, different tissue collection methods, and with varying tumor purities (Fig. 1c; Supplementary Table 1). Each was analyzed with a custom 186-amplicon panel covering 93 frequently mutated genes in PDAC (Supplementary Table 2). The mean library yield of single nuclei that passed quality control ("Methods") was 2867 complete nuclei (standard deviation = 1672.67). For context, two previous studies[17,19] using primary acute myeloid leukemia (AML) cell suspensions for Tapestri resulted in on average 5072 complete cells/sample (146 samples) and 6102 complete cells/sample (154 samples) in the final libraries.

We next determined the extent to which extracted single nuclei could be stored in suspension without affecting the yield. For two different samples (PA04-2 and PA04-3) we cryopreserved a portion of the extracted nuclei ("Methods") then thawed these frozen nuclei suspensions after 3 weeks and 14 weeks respectively as input for snDNA-seq. This allowed us to compare both the nuclei morphology and the resulting snDNA-seq results between freshly extracted and cryopreserved nuclei of the same biological samples. We found that the freeze-thaw-resuspension process had >80% recovery rate and minimal change in nuclei morphology and clumping % (Supplementary Fig. 1a, b). Moreover, the final library yield did not decrease when prepared with frozen nuclei; on the contrary, frozen nuclei gave slightly higher yield than fresh nuclei for both samples (Supplementary Fig. 1c). Frozen nuclei generated similar quality results as freshly extracted nuclei as measured by sharing the majority of high-quality variants ("Methods") (Supplementary Fig. 1d, e), and having largely linearly correlated pseudobulk variant allele frequency (VAF) for all shared variants (Fig. 1d; Supplementary Fig. 1f). For both samples, fresh and frozen nuclei revealed highly concordant genotypes as well (Fig. 1e; Supplementary Fig. 1g).

While the doublet rate inherent to the Tapestri scDNA library preparation method has been estimated to be 5–8% by its manufacturer[23], we sought to estimate the multiplet rate associated with our entire workflow when using snap-frozen tissue. We selected two samples from two distinct patients, each with a tumor population characterized by a distinct driver mutation (*TP53* p.C207Y vs. *ARID1A* splice) that was orthogonally validated by bulk WES as well as independent Tapestri runs. Similar sized pieces of each tissue sample were mixed together and subjected to the entire nuclei extraction workflow followed by snDNA library generation. Next, we assigned each cell to the two originating tumors based on its genotype for the two driver mutations. The number of barcodes that carried somatic variants from both tumors are as shown in the Venn Diagrams (Fig. 1f). Using a mixture model (Supplementary Methods) we derived the doublet rate to be 3–5%.

As a PCR-based method, our snDNA library preparation is subject to technical noise introduced by uneven amplification of both alleles in a cell (allelic dropout, ADO). To quantify the ADO rate inherent to the workflow, we selected 19 of our samples which came from resection of early-stage PDAC. We identified germline single-nucleotide polymorphisms (SNPs) by comparing single-nuclei sequencing results with bulk-sequencing results of matched normal sample of the same patient, and calculated the mean ADO rate to be 19.6% (of 100 nuclei, 19.6 nuclei only have one of the two alleles) ("Methods") (Supplementary Table 3). However, we deem this as overestimation as we observed obvious single-nucleus colocalization pattern of ADO of SNPs on different chromosome arms, which more likely suggests real copy number variation rather than technical noise.

To determine the optimal sequencing depth (in this case, mean reads per cell per amplicon) required to extract significant insights from the snDNA-seq data, we selected two samples PA04-2-frozen and PR05-3 that we deemed as over-sequenced (respectively, 278 million and 341 million total read pairs; 168 and 342 mean reads/cell/amplicon) and performed downsampling experiments by subsampling the raw FASTQ files at intervals of 50 millions read pairs (Supplementary Fig. 2a, b; Supplementary Table 4). The elbow point could be spotted at roughly 100 mean reads per cell per amplicon for the proportion of DNA read pairs assigned to cells, total number of barcodes captured, and number of cells called (Supplementary Fig. 2c–e). However, the proportion of tumor cells, as defined by the orthogonally validated *KRAS* mutation for each case, did not vary much along with total depth and mean depth per cell per amplicon (Supplementary Fig. 2f). This indicates that the relative proportions of major clones and associated biological insights could be robust to varying technical parameters such as read depth and cell throughput. However, for the purpose of rare clone discovery where cell throughput would be critical, the results here suggest at least a mean depth of 100 is required for our panel specifically.

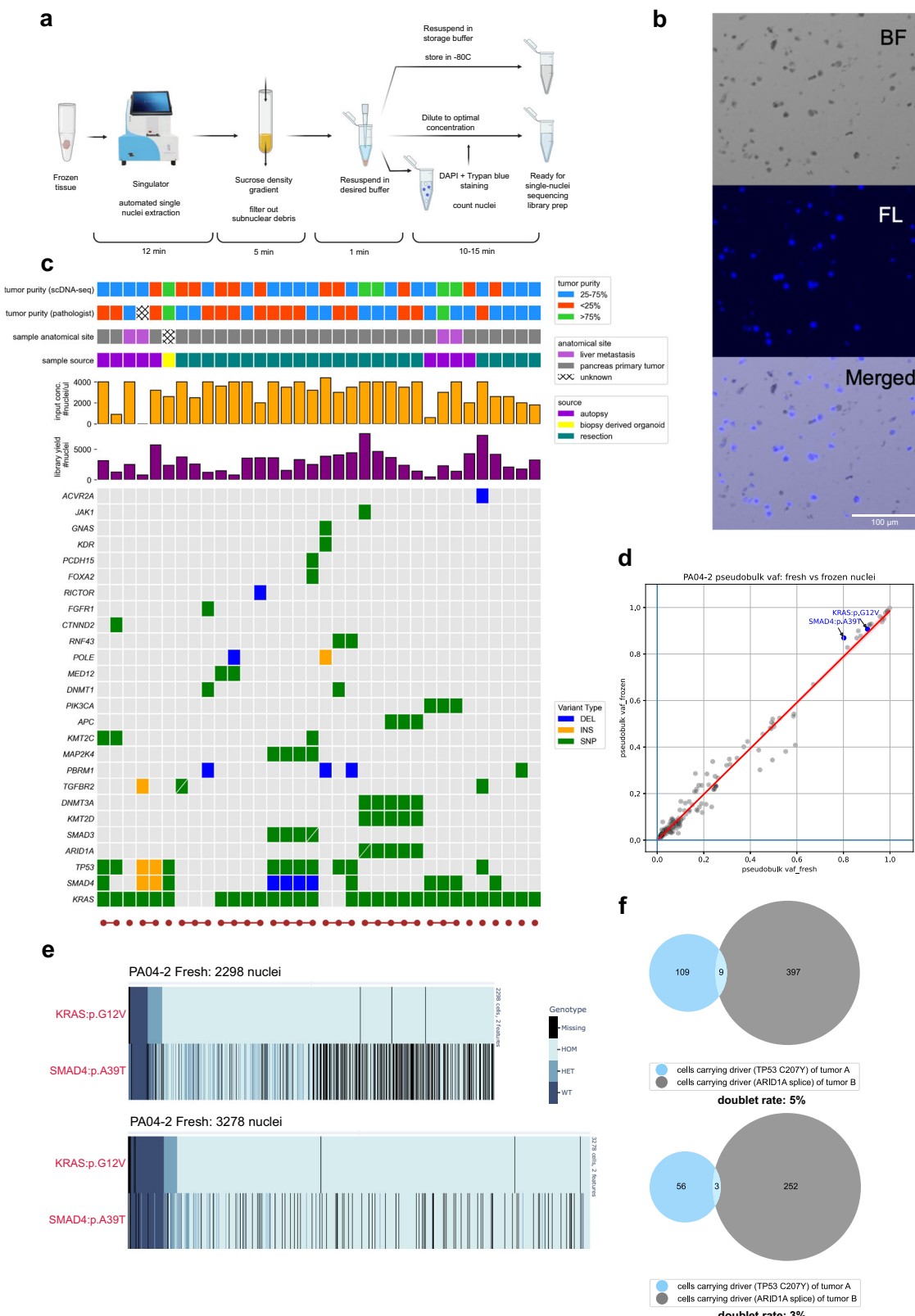

## Validation against bulk-sequencing results

To validate the robustness of our method for detection of single-nucleotide variants (SNVs), we selected 18 samples previously used for bulk WES sequencing ("Methods", Supplementary Data). When limiting to variants present within the targeted panel, a strong concordance was noted between unfiltered variants called from frozen nuclei versus those called by WES (Fig. 2a; Supplementary Fig. 4a). When we next limited to filtered, higher-confidence variants from bulk ("Methods"), there was virtually complete concordance (Fig. 2b). Moreover, despite technical differences inherent to WES versus Tapestri (input tissue slice, DNA library prep technology and sequencing depths) the bulk WES VAFs and snDNA-seq pseudobulk VAFs for all shared variants, except for those with ≤2 alternative reads in bulk WES results, were linearly correlated for the majority of samples (Supplementary Fig. 4b–d).

**Fig. 1 | Frozen tissue single nuclei extraction workflow for snDNA-seq.**
**a** Overview of frozen sample single nuclei extraction workflow for Tapestri snDNA-seq. **b** Representative microscopic images of extracted single nuclei, stained with Trypan blue (brightfield, BF), DAPI (fluorescence, FL). At least 3 representative pictures were taken per sample and yielded similar results. **c** Technical and genetic profile of each biologically distinct sample. A total of 38 samples were processed, 34 of which were biologically distinct samples from 16 unique patients. Genetic profiles were based on bulk sequencing. **d** Pseudobulk (p-bulk) variant allele frequency (VAF) comparison of all 160 shared variants between libraries prepared with fresh vs frozen (3 weeks) nuclei of sample PA04-2. Key drivers pre-identified by bulk WES in this case are highlighted; regression line with 90% confidence interval is drawn. **e** Single-cell genotype (HOM homozygous mutation, HET heterozygous mutation, WT wildtype) heatmap of snDNA-seq libraries generated by fresh vs frozen nuclei of sample PR04-2. Each row represents a bulk data-validated driver variant of this case, while each column represents a single nucleus in the library. The nuclei were sorted based on *KRAS* variant's VAF in ascending order from left to right. **f** Venn diagrams showing colocalization of genotypes belonging to two separate tumor cell populations in one snDNA-seq library (two replicates shown); the cells carrying both genotypes were identified as doublets. Nuclei suspension extracted from two tumor samples from different patients were mixed and subject to snDNA-seq. The two distinct tumor populations were identified by genotype for their respective driver variants *TP53* p.C207Y and *ARID1A* splice. Source data are provided as a Source Data file.

We next sought to determine if our snDNA-seq approach can detect subclonal copy number variations (CNVs). For this analysis we used sample PA02-1 with known homozygous deletions of *SMAD4* and *CDKN2A*[24]. By calculating the single-cell per-amplicon (~200 bp) ploidy based on read counts ("Methods"), we were able to identify both homozygous deletions (ploidy-0) in the neoplastic population despite it comprising 28.5% of all cells (Fig. 2c, d; Supplementary Fig. 5a–d).

### snDNA-seq is applicable to low-cellularity, low-tumor content samples

Low cellularity and tumor content in certain clinical settings, such as with fine needle aspirations, core needle biopsies or for PDAC in general[25], represent major hurdles for bulk-sequencing and hinder the quality of resulting genomic information. We therefore tested our workflow's performance in such settings.

We identified one sample PA04-1 of a primary pancreas tumor with both low-tumor cellularity (neoplastic cells occupying <50% area of tissue section) and low overall cellularity due to high fat content (Fig. 3a). Despite use of a tissue sample of similar size (~8 mm³) as others studied, we extracted a total of 30,000 nuclei, much lower than the optimal number of 200,000 as input for Tapestri. Ultimately, we captured only 479 nuclei in the final library, approximately 6-fold lower than average. Nonetheless, both driver gene variants (*KRAS* p.G12V, *SMAD4* p.A39T) identified by bulk WES of this same sample were identified with high-quality read data (Supplementary Fig. 6a, b) and indicated 113 likely tumor cells (23.6% of all) captured for this sample based on the presence of the clonal *KRAS* variant (Fig. 3b, c).

With a second sample (PR04-1) of particularly low-tumor purity as revealed by pathology review (Fig. 3d) and bulk sequencing (Fig. 3e), Tapestri data identified 40 out of 3866 nuclei (1%) of this sample carrying at least one of *KRAS* p.G12V, *TP53* p.R186H, *or SMAD4* p.508D (Fig. 3f, g). Again, the driver variants were genotyped with high-quality read data (Supplementary Fig. 6c, d). Intriguingly, the single-nucleus colocalization of the three main drivers violated the assumption of the infinite sites model: more than half of the nuclei carry the *KRAS* mutation and among them, a subset carry *SMAD4* and *TP53*, yet a significant number of nuclei were wildtype for *KRAS* yet mutated for *SMAD4/TP53*. If it were assumed that *KRAS* were mutated first and *SMAD4* and *TP53* mutations followed, a likely explanation would be that a subset of tumor cells lost their mutant *KRAS* allele through loss of heterozygosity (LOH). Although ADO might factor in, similar patterns observed in snDNA-seq results of two other samples of this case (PR04-2, PR04-3) seemed to support the abovementioned theory (Supplementary Fig. 6e, f). However, more rigorous statistical modeling is required for validation.

### snDNA-seq identified two mutually exclusive clones bearing two different KRAS mutations in the same PDAC patient

For a PDAC surgical resection case PR02, based on both MSK-IMPACT sequencing (high-depth targeted sequencing) and bulk WES, we identified that the major tumor clone carried the hotspot *KRAS*

p.G12D mutation; hints of a minor *KRAS* p.G12V clone existed but were on the borderline of the technologies' detection sensitivity (Fig. 4a). Single-nucleus genotype heatmaps and Venn diagrams (Fig. 4b–e) of multiregional samples PR02-3 and PR02-4 suggested colocalization of the major *KRAS* p.G12D with another likely driver *TP53* p.C203Y, which signified the major tumor clone in this sample; the minor *KRAS* p.G12V -bearing clone was mutually exclusive with the above two drivers and did not colocalize with any known driver gene mutation at similar clonal frequency. In PR02-4, while the major clone consisted of 221 cells (6.12% of all cells), the minor clone was only 12 cells (0.33% of all cells, Fig. 4e), further buttressing the technology's sensitivity. The minor *KRAS* p.G12V clone in both samples was substantiated with high-quality read data (Supplementary Fig. 7a, b); digital droplet PCR (ddPCR) also proved the presence of the *KRAS* p.G12V (Supplementary Fig. 7c). Pathology review did not identify any apparent secondary neoplastic or metaplastic structure (Fig. 4f). This observation aligns with several other studies[6,26] in suggesting that multiple *KRAS* genetic variants may coexist in one patient's PDAC precursor/tumor.

### snDNA-seq identified complex clonal structures in a KRAS-WT PDAC

The *KRAS* gene is mutated in >90% of all PDAC's and signifies the phenotype of MAPK-ERK pathway hyperactivation[25]. By bulk exome sequencing of three spatially distinct samples of resected PDAC PR01, we noted that this case was wild type for *KRAS* yet contained an *FGFR1* p.T50K mutation as well as two distinct *TGFBR2* mutations (p.M450I, p.A451G) on the same allele. Pathology review of the samples' H&E slides identified two well-isolated populations of PDAC cells with distinct histological features. A population of PDAC cells characterized as dilated glands with extensive stroma was exclusively present in sample PR01-1 (Fig. 5a, left), while another population characterized as small nests of tumor cells was exclusively present in PR01-2 (Fig. 5a, right). Sample PR01-3 had both populations present in the same tissue section. We performed snDNA-seq of all 3 samples and discovered that the *FGFR1* and *TGFBR2* mutations corresponded to two mutually exclusive clones: sample PR01-1 had only the *TGFBR2* double-mutated clone (Supplementary Fig. 8a), PR01-2 only the *FGFR1* mutated (Supplementary Fig. 8b), while PR01-3 contained both clones that were mutually exclusive at the single-cell level (Fig. 5b, c, Supplementary Fig. 8c). To determine the extent to which these two clones were unique neoplasms versus subclones that shared a common ancestor, we included all other high-quality coding & non-germline variants to define a putative normal cell population (Supplementary Fig. 8c, blue clone) and computed the median per-amplicon ploidy within each clone ("Methods"). This revealed many shared CNVs between the two clones that included allelic losses of *ARID1A, TGFBR2, FGFR1* and *SMAD4* as well as a homozygous deletion of *CDKN2A* (Fig. 5d, Supplementary Fig. 8c). Collectively these data suggest that large-scale copy number aberrations preceded the formation of the two distinct SNV clones and was likely the upstream oncogenic event.

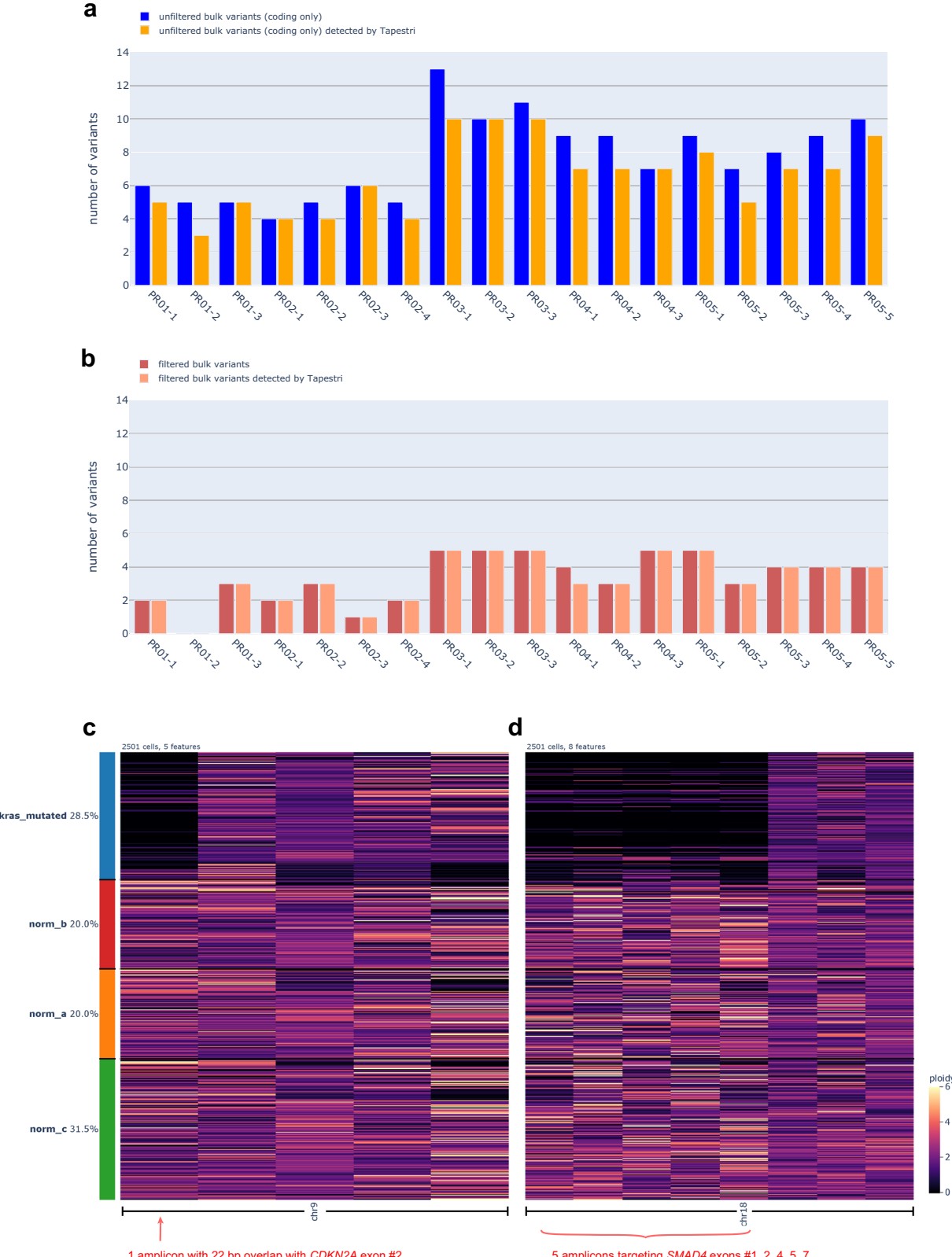

**Fig. 2 | SnDNA-seq can capture SNV and CNV pre-identified by bulk-sequencing with high sensitivity. a** For 18 samples of the same bulk WES sequencing cohort, histogram showing the number of unfiltered ("Methods"), coding variants called by bulk (blue) and among them, the number detected by snDNA-seq (orange). **b** For the same samples, histogram showing the number of filtered ("Methods") variants called by bulk (dark red) and among them, the number detected by snDNA-seq (light red). Single-cell per-amplicon ploidy heatmap of select amplicons covering chromosomes 9 (**c**) and 18 (**d**) of sample PA02-1. Each row represents one cell while each column is one amplicon. Cells are divided into *KRAS*-mutated group (blue) and 3 normal groups (red, yellow, green) ("Methods") and hierarchically clustered within group. The amplicons spanning *CDKN2A* and *SMAD4* genes are labeled. Source data are provided as a Source Data file.

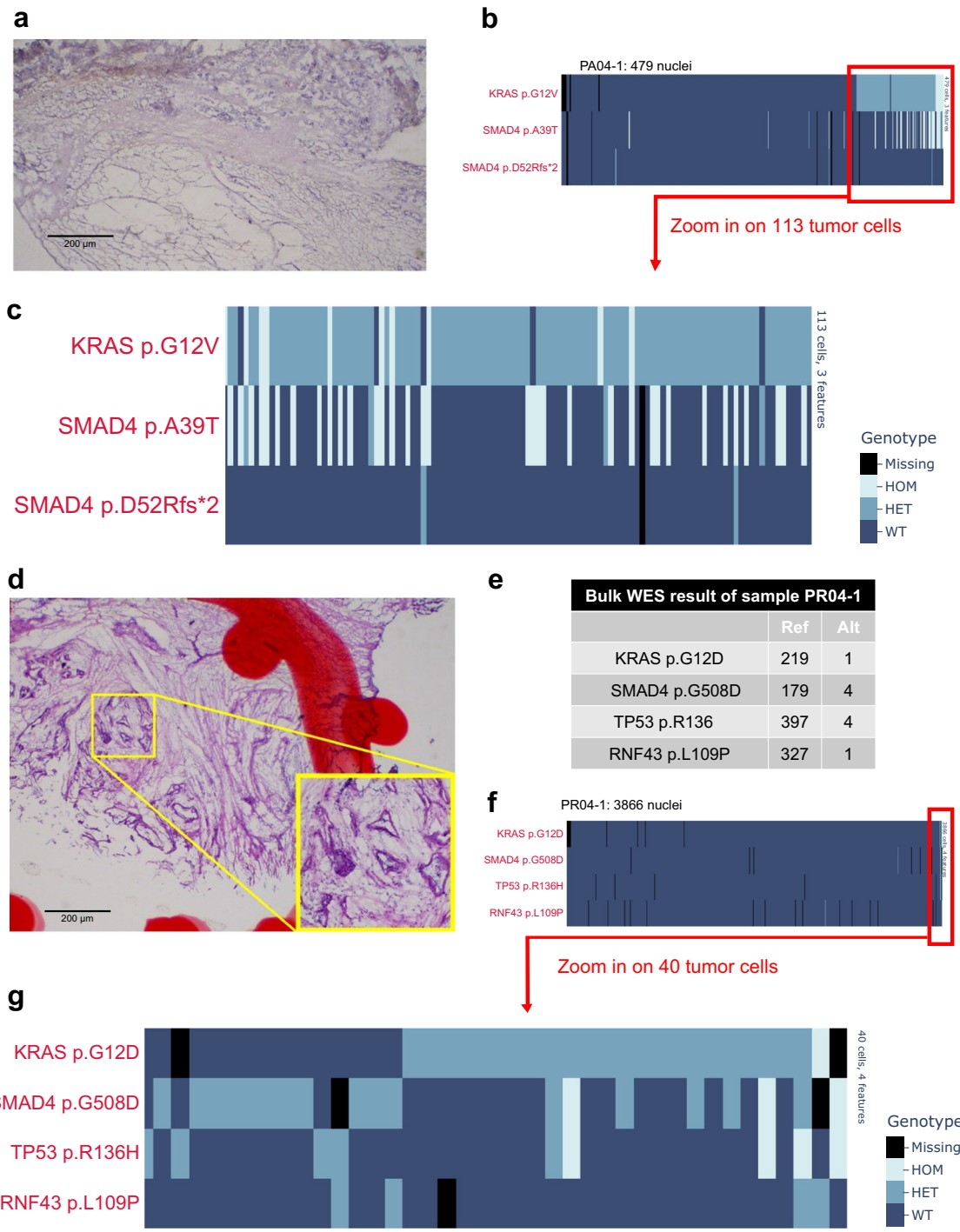

**Fig. 3 | SnDNA-seq's performance in limiting settings. a** Representative hematoxylin and eosin (H&E) stained histology image of sample PA04-1, a sample of particularly low cellularity due to high fat content. At least 3 representative pictures were taken per sample and yielded similar results. **b** Single-cell genotype (HOM: homozygous mutation; HET: heterozygous mutation; WT: wildtype) heatmap of sample PA04-1. A total of 479 captured single nuclei were sorted based on *KRAS* VAF in ascending order from left to right. **c** Single-cell genotype (HOM: homozygous mutation; HET heterozygous mutation, WT wildtype) heatmap of sample PA04-1, zoomed in on 113 putative tumor cells, defined by the presence of *KRAS*

p.G12V/*SMAD4* p.39T mutation. **d** Representative H&E histology image of sample PR04-1. At least 3 representative pictures were taken per sample and yielded similar results. **e** Bulk WES result of sample PR04-1. **f** Single-cell genotype heatmap of PR04-1. A total of 3866 captured single nuclei are sorted based on *KRAS* VAF in ascending order from left to right. Tumor cells, defined by the presence of *KRAS* p.G12D/*SMAD4* p.G508D/*TP53* p.R136H mutations, cluster to the right and are barely visible. **g** Single-cell genotype heatmap of PR04-1, zoomed in on 40 putative tumor cells.

## snDNA-seq revealed stepwise evolution during PDAC metastasis

Through snDNA-seq of three multiregional samples of PDAC autopsy case PA04 (one primary tumor sample, two liver metastases; the two liver metastases each has two technical replicates) we identified sequential steps leading to TGF-β inactivation in association with cancer progression. We started from observing the raw single-nucleus SNV genotype and per-amplicon read count data (Supplementary Figs. 9, 10), where we noticed the gene *SMAD4* underwent several steps of genomic evolution - two somatic SNV events and several asynchronous focal deletion events, as represented by clone 2 and clone 4

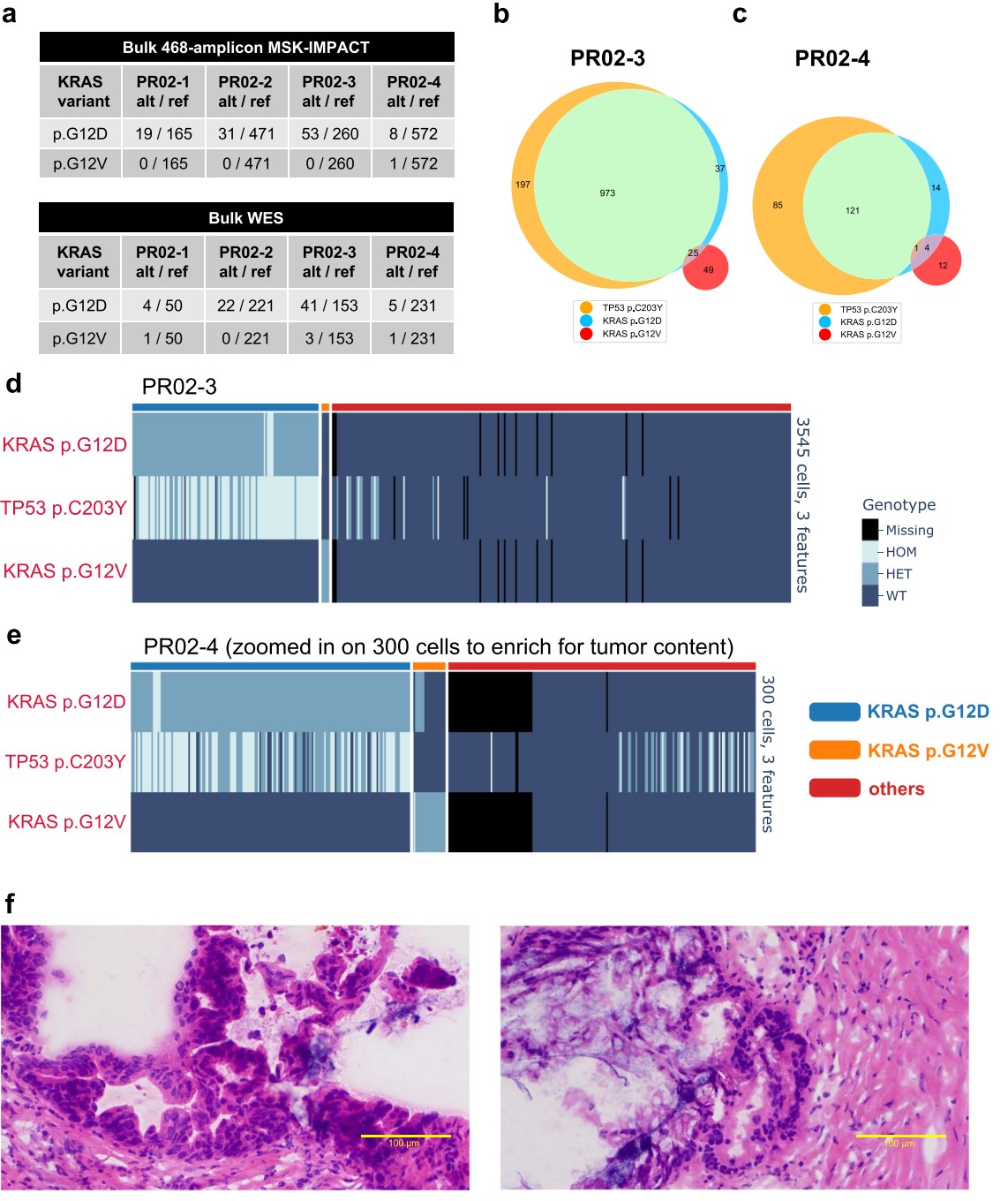

**Fig. 4 | SnDNA-seq identified two mutually exclusive clones bearing two different KRAS mutations in one pancreatic cancer patient. a** Bulk-sequencing calls of *KRAS* variants of patient PR02's 4 multiregional primary tumor samples. Venn diagram showing colocalization pattern of genetic variants *TP53* p.C203Y, *KRAS* p.G12D, *KRAS* p.G12V in single nucleus of samples PR02-3 (**b**), PR02-4 (**c**). Single-cell genotype (HOM homozygous mutation, HET heterozygous mutation, WT wildtype) heatmap of samples PR02-3 (**d**), PR02-4 (**e**, zoomed in on 300 cells where tumor

cells cluster). The *KRAS* p.G12D & p.G12V clone identities (above heatmaps) are identified as cells having "HET" or "HOM" genotype of each variant and are colored as labeled. Cells are hierarchically clustered based on the two KRAS variants' single-cell AF. *KRAS* p.G12V clone size was 57 cells in PR02-3, 17 cells in PR02-4.
**f** Representative H&E histology images of sample PR02-3. At least 3 representative pictures were taken per sample and yielded similar results.

in Supplementary Fig. 9 – which eventually resulted in clone 5 with homozygous deletion (homdel) of all *SMAD4*'s genomic region that our panel covers. Therefore, we combined the homdel (ploidy=0) status of *SMAD4*'s 8 amplicons with the genotype of 25 SNVs validated by bulk sequencing, either germline or somatic, across 8061 single cells of all three samples and applied a previously described single-cell multi-layer, multi-state clustering method[27] (Supplementary Methods). Seven clusters were identified (Fig. 6a) and the inferred phylogeny (Fig. 6b) generally aligned with the most popular model of PDAC

progression: driver SNV events occur first, large-scale CNV events follow, and focal CNV events continue as tumor cells metastasize[9]. The presence of the intermediate clusters 5 and 6 where *SMAD4* was only partially deleted compared to its end state (homdel on 7 of 8 amplicons) in clusters 0 and 3 validated our observation of stepwise focal deletion to *SMAD4*'s genomic regions (Fig. 6a, b). Absence of these intermediate clones in the primary tumor (Fig. 6c) could also indicate that these intermediate clones were selected out by the environment at the primary site or the more evolved clone 0 reseeded from the

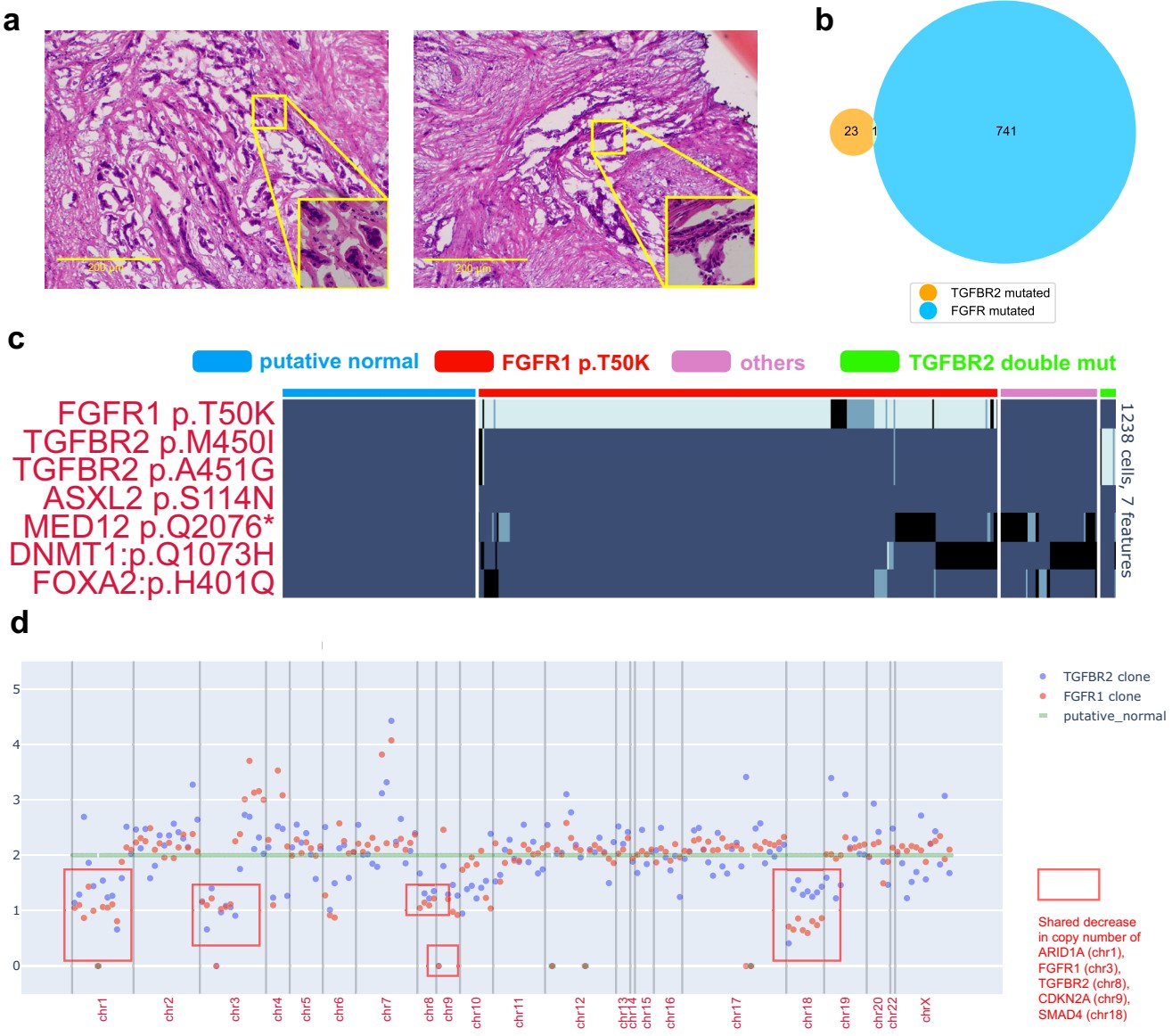

**Fig. 5 | SnDNA-seq identified two mutually exclusive SNV clones in a PDAC patient without *KRAS* mutation. a** Representative H&E histology images of PDAC regions in samples PR01-1 (left, only the *TGFBR2* double mutation was present), PR01-2 (right, only the *FGFR1* mutation was present). At least 2 representative pictures were taken per sample and yielded similar results. **b** For sample PR01-3 (both the *TGFBR2* double mutation and *FGFR1* mutation were present), Venn diagram showing single-cell colocalization pattern of the *TGFBR2* double mutation and *FGFR1* mutation. **c** Single-cell genotype (HOM homozygous mutation, HET heterozygous mutation, WT wildtype) heatmap of 7 important variants pre-identified by bulk WES for sample PR01-3. Each cell's clone identity (color strip above heatmap) is colored as shown by the figure legend. The *FGFR1* and *TGFBR2* SNV clones are defined as cells with non-wildtype genotype of each gene. The "putative normal" clone is defined as cells with "WT" genotype of all 7 genetic variants. Cells are hierarchically clustered within each clone. **d** Median per-amplicon ploidy of the *TGFBR2* double mutation clone, the *FGFR1* mutation clone and the putative normal clone found PR01-3. The putative normal clone is set as diploid baseline (green); amplicons with notable copy number loss and their corresponding genes are labeled. Source data are provided as a Source Data file.

metastasis sites; it could also be owing to the lower number of single nuclei sampled which made the smaller clones fall below the sensitivity of SCG. Interestingly, clone 5, which diverged from the main lineage (as assessed by clone size), was characterized by focal homdel of *RREB1*, which is another important effector of the TGF-β-SMAD axis[28]. Unfortunately, we were unable to resolve two minor yet important clones identified by observing the raw data – the one with a second *SMAD4* SNV and the one with homdel of all 8 amplicons of *SMAD4* (Supplementary Fig. 9, clone 4 and clone 5) – likely because their sizes were too small for SCG to differentiate them from noise. Nonetheless, these results together are in keeping with inactivation of cell-intrinsic TGFβ

signaling as a critical aspect of PDAC metastasis[29–32], as well as ongoing clonal selection for survival benefits[25,33].

## Discussion

In this study, we used two commercially available products to assemble a highly automated workflow to generate Tapestri snDNA-seq libraries from snap-frozen patient tissues. The workflow is fast, efficient and can be applicable for high-throughput clinical and translational research. Additionally, we recognized the value of storing excess single-nuclei suspensions for later use and verified a corresponding workflow that was free from issues such as nuclei quantity loss, nuclear

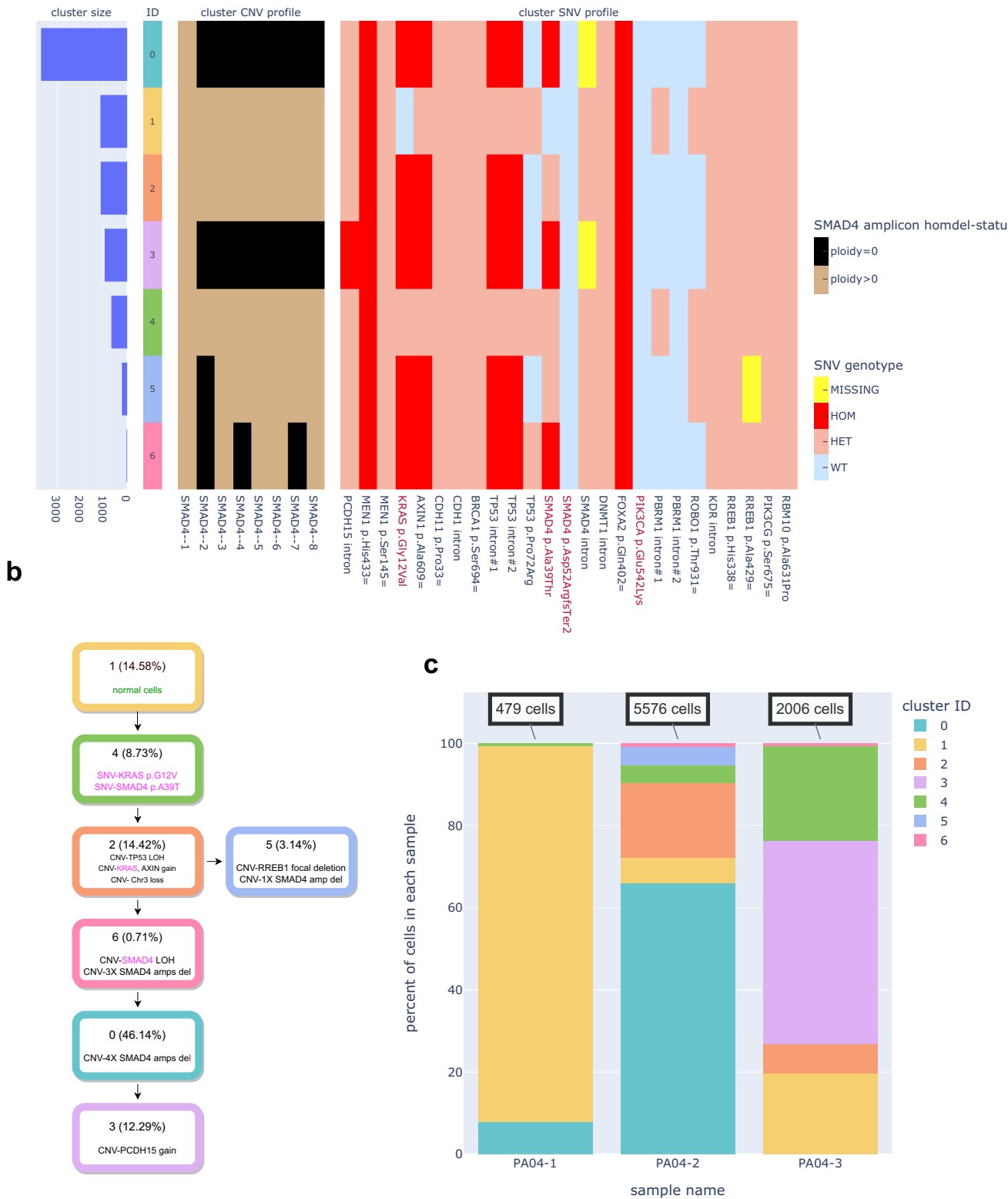

envelope damage, or nuclei clumping. This workflow further illustrates the ability to maintain information pertaining to relative VAFs compared to matched WES data. Furthermore, while our custom panel was not designed for identification of CNVs, specifically homozygous deletions, we demonstrate the proof of principle that these events could be identified with confidence. These encouraging results,

combined with the added information gleaned pertaining to co-occurring and mutually exclusive genetic events, suggests this technology and workflow is ideally suited to settings in which samples sizes are small or limited.

A caveat of this snDNA-seq technology is that albeit its high cell throughput, it is a targeted approach that focuses on a pre-designed

**Fig. 6 | SnDNA-seq revealed stepwise evolution in a metastatic PDAC case.**
**a** Single-cell genotyper (SCG)-inferred cluster information. Left panel: histogram showing each cluster's size (number of nuclei) in the 3-sample cohort. Clusters are sorted based on size from top to bottom and assigned random colors to maximize heterogeneity. Middle panel: cluster CNV profile, represented by the homozygous deletion (homdel) status of 8 *SMAD4* amplicons. Right: cluster SNV profile, represented by the genotype of 25 bulk-validated germline and somatic mutations. The 4 somatic mutations (identified by bulk sequencing on the same tumor sample) are colored red. **b** Cluster/clone phylogeny. Each cluster's proportion in the cohort is labeled next to cluster ID. Each cluster-defining genomic event is labeled on its corresponding cluster. SNV events, as well as CNV events that affect them, are colored pink. **c** Cluster composition in each of the three multiregional autopsy samples of PA04. Technical replicates (fresh and frozen nuclei preparation) are merged. Source data are provided as a Source Data file.

panel of genes which might be insufficient for certain research questions that require unbiased study of the cancer exome/genome[34–37]. This also implies a general challenge of single-cell research: because the total library size is limited by sequencing cost and data processing capacities, there is always a trade-off between the total cell throughput and the amount of information one can extract from each single cell.

Sequencing depth, usually defined as the mean number of times any nucleotide is read, is a straightforward and critical metric for bulk sequencing- it can be easily controlled by adjusting the total sequencing depth of a sample and decides the accuracy of estimating relative clonal composition of SNVs based on variant allele frequencies (VAFs) as well as the power of discovering rare SNV clones. Similarly, for Tapestri we calculated depth as the mean number of reads per amplicon per cell and determined the optimal depth. However, because reads need to be first demultiplexed and assigned to individual cell barcodes, and then undergo barcode quality control (QC or "cell-calling", "Methods"), the depth is affected by not only the total read depth, but also the number of single cells that pass QC and the proportion of reads assigned to these cells, which could be panel- and sample-specific. For example, at the same total read depth, a sample with more incomplete cells/nuclei (e.g. autopsy samples with high necrosis rate) would have a lower percentage of reads assigned to quality cells and therefore lower depth. The among-sample variation can be easily observed in the "pipeline run metadata" column of Supplementary Table 1. Therefore, it is difficult to precisely control depth through altering the total depth; it might be advisable to sequence to a total read depth that is economically feasible first, inspect the data, and sequence more if needed.

Another caveat, in this instance related to our computational analysis, is that the current variant calling pipeline for Tapestri applied GATK HaplotypeCaller and enabled default read downsampling which might be suboptimal for such high-depth single-cell data in a cancer setting. We did not apply matched normal/panel of normal (PON)-aided filtering in our pipeline; a benefit of this approach is that it enabled us to see many germline variants that can be used for quality control or phylogeny modeling, although this benefit is balanced by outputs containing a large number of artifacts. Ultimately, to enable novel somatic variant discovery a more robust variant calling pipeline is needed to adapt to the noise profiles particular to this PCR-based high-throughput single-cell DNA sequencing technology.

The field of single-cell transcriptomics began earlier than single cell genomics, and accompanying analysis methods have been flourishing in the past 5 years[38,39]. Single-cell transcriptome-oriented methods, such as clustering or gene-set enrichment analysis, are generally not optimal for targeted single-cell genomic data, where the cell-cell difference is much smaller, often only on a small set of genomic regions. The best method would be to rely on the ancestor-descendent relationship between every pair of cells' genomic sequence and build single-cell phylogenies. Several models for the evolution of SNVs in cancer have been developed to date[40–46], but only two[45,46] have been constructed to also account for the frequent, complex aneuploidy in cancer, one of which is directly applicable to the dataset presented in this paper[46] but its accuracy is yet to be tested. From the combined SNV and CNV data of case PR01's multiregional samples (Supplementary Fig. 8a–c), we saw many complex clonal structures than could be defined by driver SNV events alone. Clustering single cells based on both SNV and CNV in PA04's multiregional

samples, we were able to resolve stepwise evolution patterns in metastatic PDAC, showing single-cell-level convergent evolution to inactivating the TGF-β pathway, although at a limited resolution. Therefore, better methods need to be developed and tested to analyze this new dataset further.

## Methods
### Ethics statement
Use of samples used in this study was approved by the institutional review board at Memorial Sloan Kettering Cancer Center (under protocols #15-149, #15-021) and Johns Hopkins Medicine.

### Patient sample collection and preprocessing
Patient samples used in this study mainly consist of two categories-multiregionally sampled surgical resection of primary pancreatic cancer, and multiregionally sampled autopsy of metastatic pancreatic cancer. Each participating patient provided written consent and was not compensated. Detailed patient and sample information is summarized in Supplementary Table 1.

For multiregional sampled surgical resections: treatment naïve patients with tumors ≥2 cm on cross-sectional imaging were identified preoperatively. A single cross-sectional piece of tumor was sampled sequentially using a cartesian coordinate system with 0.6 cm × 0.6 cm grid, with 3–5 samples obtained from each tumor. Adjacent normal pancreas or duodenum was also collected. All samples were stored at −80 °C until use.

For multiregional sampled autopsy: Tissues from three patients were used. All patients had a premortem diagnosis of PDAC based on pathological review of resected biopsy material and/or radiographic and biomarker studies.

Tissue sections were cut from tissue blocks embedded in optimal cutting temperature (OCT) compound, stained with hematoxylin and eosin (H&E) and reviewed by a gastrointestinal pathologist (S.U.) to estimate total cellularity, tumor purity and tissue quality. Specifically, tumor purity was estimated based on the ratio of tumor cell-occupying 2-D area against all area, and therefore might not conform with that estimated from snDNA-seq result, which was defined as the ratio of tumor nuclei (defined as carrying known driver SNVs from bulk sequencing data) of all nuclei. Normal samples were reviewed to confirm that no contaminating cancer cells were present.

### Bulk WES, WGS library preparation, sequencing, and variant calling
Genomic DNA was extracted from each tissue using the phenol-chloroform extraction protocol or QIAamp DNA Mini Kits (Qiagen). WGS, WES and alignment were performed by the Integrated Genomics Operation and the Bioinformatics Core at Memorial Sloan Kettering Cancer (New York, NY). Briefly, an Illumina HiSeq 2000, HiSeq 2500, HiSeq 4000 or NovaSeq 6000 platform was used to target sequencing coverages of >60× for WGS samples and >150× for WES samples.

Sequencing reads were analyzed in silico to assess quality, coverage, as well as alignment to the human reference genome (hg19) using BWA. After read de-duplication, base quality recalibration and multiple sequence realignment were completed with the PICARD Suite and GATK v.3.1; somatic single-nucleotide variants and insertion−deletion mutations were detected using Mutect v.1.1.6 and HaplotypeCaller v.2.4. Such a process generates the "filtered" variant

list for every sample. Then, all variants of all samples of the same sequencing cohort were pooled as a single list. Each sample's BAM file were used to compute "fillout" values (total depth, reference allele counts, alternative allele counts) for each variant in the pooled list. An alternate read > 2 filter was applied to trim down false positives. This process aimed to rescue variants that were detected with high confidence in multiregional sample #1 but with low confidence in multiregional sample #2 of the same patient; the output corresponded to the "unfiltered" variant list.

## Nuclei extraction from frozen tissue, counting, QC, sorting and cryopreservation

Single nuclei from OCT-embedded snap-frozen primary tissue samples were extracted using the Singulator 100 machine (S2 Genomics) with its extended nuclei dissociation protocol. After extraction, nuclei solution was centrifuged at $800 \times g$ for 5 min in a swing bucket with a reduced braking in a 0.25 M Nuclei PURE Sucrose solution (Sigma-Aldrich) to filter out debris.

Nuclei were stained with Trypan blue and manually inspected under a brightfield microscope for clumping percentage, which was estimated as the number of clumped particles out of all single particles within one field of view. Nuclei concentration was estimated by DAPI staining on a Countess II FL automated cell counter. Clumping percentage and nuclei concentration were both measured $\geq 2$ times for each sample. A final concentration of 4000 nuclei/µl suspended in 50 µl Mission Bio cell buffer was targeted per sample prepared.

After up to 200,000 nuclei were taken for Tapestri library preparation, the remaining nuclei were resuspended in Sigma Aldrich Nuclei PURE storage buffer and immediately frozen on dry ice, before being transferred to −80C freezer for long-term storage. If needed, nuclei were thawed on ice until the solution was clear, and centrifuged with the same settings as described above for pelleting and buffer exchange.

## Single-nuclei library preparation and sequencing

Nuclei were suspended in Mission Bio cell buffer at a maximum concentration of 4000 nuclei/µl, encapsulated in Tapestri microfluidics cartridge, lysed and barcoded. Barcoded samples were then put through targeted PCR amplification with a custom 186-amplicon panel covering important PDAC mutational hotspots in our sample cohort (Supplementary Table 2).

The 186-amplicon panel was designed based on curation of bulk whole exome/genome sequencing data of PDAC samples collected by the Iacobuzio lab. The goal was to cover as many likely driver SNVs within our patient cohort as possible within a 200-amplicon limit, which we set considering the economic cost and the purpose being mostly proof of-principle for a variety of ongoing projects in our lab. The driver SNVs/genes of interest were determined by querying several public databases including OncoKB, Cancer Genome Interpreter (CGI), cancer hotspots[47], TCGA consensus driver gene list[48] etc. using Treeomics[49] and LiFD[50] that the Iacobuzio lab helped develop for previous studies. In addition to the driver SNVs from above, we used cBioportal[51,52] to query additional frequently amplified genes in PDAC (e.g. *MYC*) and added coverage for them as well.

PCR products were removed from individual droplets, purified with Ampure XP beads and used as templates for PCR to incorporate Illumina i5/i7 indices. PCR products were purified again, quantified with an Agilent Bioanlyzer for quality control, and sequenced on an Illumina NovaSeq. The minimum total read depth was determined by the formula:

$$X = (\text{expected number of nuclei called}) \times (\text{number of amplicons}) \\ \times (\text{target depth}) \div (\text{expected proportion of reads assigned to cells}) \quad (1)$$

We estimated expected number of nuclei called as (input nuclei concentration) * 1 µl (e.g. an input suspension of 4000 nuclei/µl will yield 4000 nuclei). The target depth was set at 100 reads per cell per amplicon. The expected percentage of reads assigned to cells was set at 0.5. However, due to parallelizing multiple samples on the same lane, the total read depth could not be precisely controlled and deviated from target for a few samples.

## Single-nuclei DNA library quality control, cell-calling and variant calling

FASTQ files for single-nuclei DNA libraries were processed through Mission Bio's Tapestri pipeline with default parameters. Briefly, it trims adapter sequences, aligns reads to the hg19 genome (UCSC), assigns reads to cell barcodes. The CellFinder module then filtered for barcodes corresponding to "complete cells/nucleus" based on total read completeness (>8 * number of amplicons) and per-amplicon read completeness (>80% data completeness for working amplicons, which are defined as amplicons with > 0.2*mean of all amplicon reads per qualified barcode). It next used GATK HaplotypeCaller to call variants individually on each cell, and then GATK GenotypeGVCFs to jointly genotype all cells using genotype likelihoods from the previous step. The unfiltered VCF was parsed into an HDF5 file containing single-cell variant and per-amplicon read count matrices compatible with downstream analysis. A more detailed documentation of the pipeline is available at: https://support.missionbio.com/hc/en-us/categories/360002512933-Tapestri-Pipeline. In respect of Mission Bio's request, the pipeline code is not to be publicized because it contains proprietary information per industry standard. However, the pipeline used in the paper that demonstrated this scDNA-seq library preparation technology[53] is publicly available as a Github repository at https://github.com/AbateLab/DAb-seq. Although we have not formally tested that it performs identically as the Mission Bio pipeline, we believe it is sufficient to replicate our results.

## Single-cell genotyping and cell-variant pair filtering

The HDF5 file output from above was analyzed mainly by Mission Bio's python-based analysis package Mosaic, with a modified genotyping and variant filtering module. As shown in Supplementary Fig. 2, with a single-cell variant call matrix, we started by assigning a genotype to each cell-variant pair. First, we defined the minimum depth at 5 reads and any variant in any cell with depth below the threshold in a cell would be assigned as "missing". Then, we used cutoffs:

$$VAF_{WT} \sim [0, 20]$$

$$VAF_{HET} \sim (20, 80]$$

$$VAF_{HOM} \sim (80, 100]$$

to assign each variant's genotype (WT- wildtype; HET- heterozygously mutated; HOM- homozygously mutated) in each cell, thus allowing 20% of reads of a barcode to be false positives potentially caused by barcode contamination. Finally, we set the threshold for the alternate (mutant) read count to 3 reads to convert low-quality heterozygous calls back to wildtype to arrive at the final cell-variant genotype matrix. This alternate read filter was adapted from SNV filtering for bulk sequencing. It was also intended to filter out SNVs with low depth and low alternate read count, which are very likely false positives caused by misalignment or homopolymer region or droplet merging. Note that this genotyping method is intended to assign single-cell genotypes for bulk-validated SNVs but not for novel SNV discovery.

For getting a list of high-quality variants for each library de novo (to compare the SNV set between fresh and frozen nuclei), we used a more stringent variant filtering scheme than above: starting from the

genotype matrix output by Mission bio's variant calling pipeline (the default **NGT** matrix, output by the **h5.create.dna.create_ngt()** function), we first genotyped each mutation call with a more stringent filter - minimum depth was set to 10 while minimum alternate read count was set to 5. Then we discarded variants that have "missing" genotype in more than 75% cells (while whitelisting SNVs in certain genes that are known to be prone to homozygous deletion in PDAC, such as *SMAD4, CDKN2A*). Through inspecting the distributions of cellular prevalence of variants across different read depths and total cell numbers, we determined that a mutational prevalence of 0.5% (an SNV needs to have either HET or HOM genotype in greater or equal to 0.5% of all cells) is a feasible and effective cutoff to filter out most technical artifacts. Any variant mutated in more than 0.5% of all cells was added to the high-quality variant list. See more details in the **mosaic.dna.filter_variants()** function.

### Allelic dropout (ADO) calculation

To calculate ADO, we used germline single-nucleotide polymorphisms (SNPs), which should have a heterozygous genotype and therefore nonzero read count of both alleles in all nuclei detected if there were no ADO. The ADO rate was formulated as:

$$\frac{\text{number of nuclei with strictly 0 read of either allele}}{\text{total number of nuclei}} \quad (2)$$

To identify germline SNPs, we first annotated snDNA-seq's SNV callset with bulk-sequencing variant allele frequency (VAF) of the matched normal sample, if the SNV was detected in the normal by bulk. We selected SNVs with VAF > 0.2 in the bulk normal to avoid technical artifacts. Then we calculated the mean single-nucleus VAF (not considering VAF = 0) for each SNV, and selected SNVs with 0.2 <mean single-nucleus VAF < 0.8. This was to exclude SNVs that are homozygous or associated with somatic mosaicism. The final list was then used for ADO calculation.

### Doublet model and calculation

Please see attached Supplementary Methods.

### Single-cell per-amplicon ploidy calculation

The ploidy calculation was mainly based on Mission Bio's Mosaic package. The per-amplicon read counts were normalized first within the same cell across different amplicons by mean read depth, and then within the same amplicon across different cells by median read depth. Note the median read depth across different cells only considered good-quality cells, which are defined as those with at least 1/10 number of reads as that of the cell with the 10th rank in terms of read count.

Then the per-amplicon ploidy was calculated by setting a group of cells as diploid baseline based on a priori knowledge (e.g. *KRAS* mutational status) and taking the ratio of every other cell's per-amplicon read count against that group's per-amplicon median read count.

To test the robustness of our ploidy calculation, we picked one sample with known *KRAS* mutation and cancer-related aneuploidy based on bulk sequencing and validated with DNA microarray. We started by separating a snDNA-seq library into two groups- *KRAS*-mutated and *KRAS*-WT, with the latter assumed as mostly normal cells. Then we divided the normal cell population randomly into 3 groups (norm_a, norm_b, norm_c), used one group (norm_a) as diploid baseline and calculated other groups' ploidy against it. As shown in the Supplementary Fig. 3, the two other putative normal cell groups had their median per-amplicon ploidy aligning close to 2, which validates the diploid-defining rule; the *KRAS*-mutated group had apparent aneuploidy across most amplicons and *CDKN2A* and *SMAD4* loss, which validated our ploidy calculation.

For case PR01, because there was not an a priori clonal oncogenic driver such as a *KRAS* variant to be reliably used to determine a diploid population, we used seven (including two *TGFBR2* mutations that are 2-bp apart) variants pre-identified by bulk WES to set up a rule: a cell with "WT" genotype for all 7 variants can be called putative normal and be used as diploid baseline.

### Single-cell genotyper (SCG) setup

Please see attached Supplementary Methods.

### Reporting summary

Further information on research design is available in the Nature Portfolio Reporting Summary linked to this article.

## Data availability

Raw and processed sequence data have been deposited at the European Genomephenome Archive (EGA), which is hosted by the European Bioinformatics Institute and the Centre for Genomic Regulation, under accession number EGAS00001006024. These data include: 1. Tapestri data for 38 unique sample runs and 2 mixing experiment runs: (a) FASTQ files. (b) BAM files (valid cell barcodes only) output by the default Mission Bio Tapestri pipeline. (c) HDF5 files output by the default Mission Bio Tapestri pipeline. 2. Matched bulk data used in the paper for a subset of samples: (a) WES BAM files for 19 tumor samples of cases PR01 through PR05, and the matched normal sample for each case. (b) MSK-IMPACT BAM files for 4 tumor samples of case PR02. (c) WGS BAM files for case PA04's two tumor samples (PA04-1 and PA04-liver) and one matched normal sample. The above data are available under restricted access, as required by the MSKCC Medical Donation Program Data Access Agreement (MSKCC MDP DAA). Readers interested in gaining access through EGA need to contact the data access committee (DAC) of this dataset and start an application. The DAC will try to respond within two weeks but may take longer in special conditions. The MSKCC MDP DAA, to be provided by the DAC and include guidelines and restrictions on data usage, must be signed. Once the application is approved, an EGA account will be provided for data access. The time length of access to the data will be determined by the DAC on a case-by-case basis. Further information about EGA can be found at https://ega-archive.org and "The European Genomephenome Archive of human data consented for biomedical research" (http://www.nature.com/ng/journal/v47/n7/full/ng.3312.html). Additionally, the following data are included in Supplementary Data: 1. Under folder "PR01-05-bulk_data": bulk WES mutation annotation format (MAF) files for cases PR01 through PR05. Note these are already subset to our targeted panel's genomic region. 2. Under folder "PA04-bulk_data": bulk WGS MAF file for case PA04's two tumor samples. Bulk WGS mutant allele counts for genomic variants identified for case PA04's normal sample. Again, these are already subset to our panel's genomic region. 3. Under folder "PA04-SCG_input": immediate input files used for single-cell genotyper (SCG) run (Fig. 6). These were generated as described in Supplementary Methods, Section 2. Any other data can be made available upon request. Source data are provided with this paper.

## Code availability

A customized version of Mission Bio's "Mosaic" package (original: https://github.com/MissionBio/mosaic) used in this work for snDNA-seq data analysis is available in Github (https://github.com/haochenz96/mosaic) or Zenodo (https://doi.org/10.5281/zenodo.7236672)[54]. A forked version of the single-cell genotyper (SCG) package, which only reformatted the README from the original version for better readability, is available at https://github.com/haochenz96/scg.

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

## Acknowledgements
This work was funded by the NIH/NCI grant R35 (CA220508-03) and U2C (CA233284-03) awarded to C.A.I-D. Cycle for Survival for David Rubenstein Center for Pancreatic Cancer Research. The Alan and Sandra Gerry Metastasis and Tumor Ecosystems Center for R.C. and I.M. The Daiichi-Sankyo Foundation of Life Science Fellowship and the Mochida Memorial Foundation for Medical and Pharmaceutical Research Fellowship to A.H. The NIH/NCI Ruth L. Kirschstein T32 M.D./Ph.D. Training Grant (GM141938-01) for A.Z. NIH/NCI K99/R00 (CA229979-01A1) to A.M-M. and F31 (CA260796-01) to K.M. We are grateful to the Single Cell Research Initiative (SCRI), Integrated Genomics Operations (IGO), and the bioinformatics core at MSKCC for their excellent technical support. We thank the Mission Bio field scientist team for their technical support. We thank Drs. Robert Bowman and Linde Miles (Ross Levine lab at MSKCC) for sharing their experience with Tapestri. We thank Dr. Andrew Roth (University of British Columbia, Canada) for sharing insights on single-cell genotyper. The flowchart in Fig. 1a was created with BioRender.com. The sample technical and genetic profile plot was created with CoMut[55].

## Author contributions
H.Z. and C.A.I-D. designed the study; R.C., I.M., H.Z., E-R.K. optimized the frozen tissue nuclei extraction workflow; S.U., A.H., C.A.I-D. performed pathology review; E-R.K., C.A.M., A.H., A.Z., K.M. collected, processed and sequenced all primary samples; H.Z. C.A.M., A.H., K.M. analyzed bulk-sequencing data; J. H. designed the Tapestri panel; H.Z., P.S., A.M-M., C.A.I-D. analyzed snDNA-seq data; H.Z., P.S. built the doublet model and performed calculation; H.Z. and C.A.I-D. wrote the manuscript; all authors reviewed and edited the final manuscript.

## Competing interests
The authors declare no competing interests.
