## [Peer Review File · Nature Communications]

Application of high-throughput single-nucleus DNA sequencing
in pancreatic cancerREVIEWER COMMENTS

Reviewer #1 (Remarks to the Author): Expert in single-nucleus sequencing method development and droplet microfluidics

This Paper describes a study of pancreatic cancer using single nuclei sequencing. It combines two innovative technologies, the S2 genomics singulator for automated nuclei extraction and the Mission Bio tapestri for single nuclei DNA sequencing. While these kinds of nuclei extraction studies have been possible previously, the workflows required to extract nuclei are laborious and can lead to variable results. Therefore a powerful aspect of the approach described in this paper is the use of the singulator to automate this process and achieve higher reproducibility, consistency, and yield. Therefore the technology is interesting and valuable since it facilitates single cell DNA sequencing of solid tumors that require nuclei extraction. In addition, the biological findings are interesting and well done. Overall the paper is of high-quality and I recommend publication.

Reviewer #2 (Remarks to the Author): Expert in pancreatic adenocarcinoma genomics

In this manuscript, Zhang et al. optimized a highly automated nuclei extraction workflow to perform targeted single-nucleus DNA sequencing. The robustness of the method for detection of the SNVs was validated using matched bulk WES sequencing samples. This method was further demonstrated to be applicable to even low-cellularity and low-tumor content PDAC samples. Generally speaking, the study presented a useful and scientifically sound method that may be feasible and widely applicable in the field. My detailed questions are listed below.

1. Is the tumor purity consistent among pathological review, the % derived by bulk WES sequencing and % derived by snDNA-seq?
2. How was the custom 186-amplicon panel for highly mutated PDAC genes determined? How were the genes selected? Was there a reference that the authors used?
3. It may be helpful to discuss more about how the sequencing depth would affect the results shown, especially for others who wants to use the method in the future. A down-sampling analysis may be one of the choices.

Reviewer #4 (Remarks to the Author): Expert in bioinformatics, cancer single-cell genomics and evolution, and phylogenetic inference

Review of the first submission of "Application of high-throughput, high-depth, targeted single-nucleus DNA sequencing in pancreatic cancer"

SUMMARY:

The primary goal of this work is design of time efficient and cost effective workflow for single nuclei extraction from solid tumor tissues, which is then followed by library preparation and

DNA sequencing steps yielding single-nucleus DNA sequencing data that can be used for various types of analyses (e.g, variant detection, deciphering tumor subclonal composition, studying evolutionary history and others). The authors employ this workflow to prepare 38 single-nucleus DNA libraries from 34 samples collected from 16 patients diagnosed with pancreatic ductal adenocarcinoma (PDAC). On average, 2867 nuclei per sample were sequenced using targeted 186-amplicon panel designed to cover the most frequently mutated genes in this type of cancer. Similar studies of a larger scale (both in terms of the number of patients and the number of cells sequenced per patient) were published recently (e.g., Morita et al. from 2020), but their focus is on analyzing liquid malignancies where obtaining samples is typically easier than in solid tumors. It is worth noting that the authors acknowledge and refer to these related studies, but they also present arguments for the importance of their workflow presented in this work.

Due to my background, my review will mostly focus on the data analysis part of this work. I leave the assessment of the novelty, relevance and importance of some parts of this work, such as the cells extraction, to reviewers more familiar with wet lab.

Overall, I am positive about this work and recommend revision that would address my concerns raised below, except for possibly not performing additional experiments to address suggestions related to copy number aberrations detection if such experiments would be too laborious and/or require extensive additional resources.

MAJOR COMMENTS:

To start with, it seems that the github link provided in the paper is broken (<https://github.com/haochenz96/mosaic>) so I could not have a look into the codes used for performing data analysis. Consequently, I am not able to make any comments regarding them. Perhaps there is an access issue or the provided link is incorrect. I also leave the possibility that something is wrong on my end. Can you please have a look into this and make sure that the code is accessible (and well documented) for the next round of review.

Page 7, the last paragraph: why do you focus only on coding variants when assessing the concordance between the sets of variants called on bWES and snDNAseq data? Or, is it that, due to the design of the panel, any variant in the regions covered by the panel is considered to be coding by default hence mention of the word "coding"?

[minor, but I place it here due to the comment below] Page 8, I to some extent agree that detecting a homozygous deletion present in around 30% of cells might be challenging from low quality bulk sequencing data (this certainly also depends on the size of the deletion and several other factors). But, in this particular context, is it appropriate and fair to contrast single-cell data with low-depth/low-coverage bulk data? I think that more appropriate comparison would be to compare single-cell data with high quality bulk data considering resources required to generate each. Even if the costs of obtaining each are comparable, it should be kept in mind that bulk data would usually yield much broader coverage. Shortly, while I appreciate that you conduct some preliminary experiments to demonstrate potential of detecting some (subclonal) copy number aberrations from the data generated in this work, I would recommend to rewrite this part by either completely removing any mention of bulk data or slightly toning down these claims and providing fairer comparison.

Furthermore, more importantly and related to the previous comment, while homozygous

deletions represent one of the simpler-to-detect types of copy number aberrations when analyzing single-cell data, it is not clear how reliably some more challenging copy number aberrations can be detected from data generated using this pipeline (e.g., aberrations where exactly one copy of maternal and one copy of paternal allele is gained). Some analysis is shown on Pages 11, 22 and 23, but, overall, I think that the manuscript would benefit if more attention is devoted to discussing these and if some related analyses and experiments are conducted, if possible (I understand that such analyses also depend on the extent of copy number aberrations present in the collected samples and availability of alternative means of validating them). Note that I am aware that detecting copy number aberrations is challenging, especially from targeted single-cell data, so I do not necessarily expect most of the results to be positive. However, providing more comprehensive discussion of which copy number aberrations are reliably detectable from this type of data can be very useful to the potential readers of this manuscript.

Would it be possible to reconstruct and present one or a few possible trees showing clonal evolution and metastatic seeding events for the case discussed on Pages 12 and 13? Although the cohort size is very limited to study metastasis on a large scale, could you make any potentially interesting observations about the metastatic seeding in the two patients (PA03 and PA04) for which sequencing data from both, primary and metastatic site(s), were generated?

Please provide some additional clarification and motivation for the sentence "Finally, we set the threshold for the alternative (mutant) read count to 3 reads to convert low-quality heterozygous calls back to wildtype to arrive at the final cell-variant genotype matrix". Also clarify what "prevalence" means in "... mutational prevalence of 0.5 %" (previous to the last line on Page 21) (i.e., provide a clear definition for how a prevalence mentioned here is calculated).

The second paragraph of section "Single-cell per-amplicon ploidy calculation" (i.e., the paragraph starting with "Then the per-amplicon ploidy ..."). Have you made available all the information required to have this part of the analysis reproducible?

Lastly, I have one question related to Data availability: it is stated that all data supporting the findings of this study are in the process of uploading to EGA and will be available upon publication. Can you please comment on whether any researcher granted access to this dataset is expected to be able to reproduce all the presented results without having to make some payments to Mission Bio (e.g., paying to get premium access to access some Tapestry barcodes in order to be able to demultiplex sequencing data or having to buy an access to some data analysis software etc.)?

MINOR COMMENTS:

In the Doublet rate estimation section, please be more precise by saying that "... a contain only mutations from cell line A" (do analogous for b).

At some places, for example in the first two paragraphs on Page 8, it seems that snDNA-seq and scDNA-seq appear interchangeably. Can you please comment on this (not necessarily in the main manuscript but at least in response to this review)?

Although I find this paper to be well written, I noticed some minor typos and several examples of them are provided below.

- in the abstract, should it be "pancreatic ductal adenocarcinoma" at the place where abbreviation PDAC is first introduced?
- "single cell partitioning systems" -> "single-cell partitioning systems"
- "nuclei clumping %" -> "nuclei clumping percentage"

Please revisit the following:

- Page 10 "... we identified the major tumor clone carry the ..."
- Page 15 "we could already see many complex clonal structures than could not be defined by driver SNV clones alone"
- Page 17 "Detailed patient, sample information ..."
- Is comma missing after "microfluidics cartridge" (previous to the last line on Page 19)?

Salem Malikic
National Cancer Institute

Manuscript NCOMMS-22-08919

Responses to referees

Haochen Zhang^{1,2}, Christine Iacobuzio-Donahue^{2,3,4*}

¹ Gerstner Sloan Kettering Graduate School of Biomedical Sciences, Memorial Sloan Kettering Cancer Center, New York, NY, USA.

² Human Oncology and Pathogenesis Program, Memorial Sloan Kettering Cancer Center, New York, NY, USA.

³ David M. Rubenstein Center for Pancreatic Cancer Research, Memorial Sloan Kettering Cancer Center, New York, NY, USA.

⁴ Department of Pathology and Laboratory Medicine, Memorial Sloan Kettering Cancer Center, New York, NY, USA.

*Correspondence: iacobuzc@mskcc.org

We thank the editor and reviewers for their helpful feedbacks. We highlighted/commented substantial changes made to the manuscript per reviewers' comments. Below we provide a point-by-point response (**blue text**) to each reviewer's comments (**black text**). All references to sections and figures refer to the revised manuscript.

Reviewer #1:

This Paper describes a study of pancreatic cancer using single nuclei sequencing. It combines two innovative technologies, the S2 genomics singulator for automated nuclei extraction and the Mission Bio tapestri for single nuclei DNA sequencing. While these kinds of nuclei extraction studies have been possible previously, the workflows required to extract nuclei are laborious and can lead to variable results. Therefore a powerful aspect of the approach described in this paper is the use of the singulator to automate this process and achieve higher reproducibility, consistency, and yield. Therefore the technology is interesting and valuable since it facilitates single cell DNA sequencing of solid tumors that require nuclei extraction. In addition, the biological findings are interesting and well done. Overall the paper is of high-quality and I recommend publication.

We appreciate your positive feedback.

Reviewer #2:

In this manuscript, Zhang et al. optimized a highly automated nuclei extraction workflow to perform targeted single-nucleus DNA sequencing. The robustness of the method for detection of the SNVs was validated using matched bulk WES sequencing samples. This method was further demonstrated to be applicable to even low-cellularity and low-tumor content PDAC samples. Generally speaking, the study presented a useful and scientifically sound method that may be feasible and widely applicable in the field. My detailed questions are listed below.

We appreciate your positive feedback.

Question 1: Is the tumor purity consistent among pathology review, the % derived by bulk WES sequencing and % derived by snDNA-seq?

Thank you for this comment. The estimated tumor purity based on pathology review and snDNA-seq is plotted in **Figure 1c**. We apologize that the figure annotations were quite small in the originally submitted version and will pay special attention to ensure its visibility in the final publication. The purity inferred from bulk WES/WGS sequencing was not included here because the current computational method (fraction and allelic copy number estimation from tumor/normal sequencing, FACETS) is known to be error prone.

As can be seen in **Figure 1c**, the purity estimated by a pathologist does not always align with that calculated from snDNA-seq data. This is likely because:

- a) a pathologist's review process can be subjective;
- b) our pathologist (S.U.) used tumor-vs-normal 2-D area ratio to estimate tumor purity, whereas for snDNA-seq, we defined single tumor nuclei as those carrying known driver mutations, and calculated the ratio of tumor nuclei of all nuclei as the tumor purity. In the case where the tumor cells are densely populated, the area-based tumor purity would be underestimation compared with single-nuclei-based purity; in the opposite case (e.g. there are dense clusters of immune cells) the area-based tumor purity would be overestimation compared to single-nuclei-based purity.

A summary of above has also been added to the Methods section "Patient sample collection and preprocessing".

Question 2: How was the custom 186-amplicon panel for highly mutated PDAC genes determined? How were the genes selected? Was there a reference that the authors used?

Thank you for the comment and we recognize the need to elaborate on the panel design process.

The 186-amplicon panel was designed based on curation of bulk whole exome/genome sequencing data of PDAC samples collected by the Iacobuzio lab. The goal was to cover as many likely driver SNVs within our patient cohort as possible within a 200-amplicon limit, which we set considering the economic cost and the purpose being mostly proof-of-principle for a variety of ongoing projects in our lab.

The driver SNVs/genes of interest were determined by querying several public databases including OncoKB, Cancer Genome Interpreter (CGI), cancer hotspots¹, TCGA consensus driver gene list² etc. using Treeomics³ and LiFD⁴ that the Iacobuzio lab helped develop for previous studies.

In addition to the driver SNVs from above, we used cBioportal to query additional frequently amplified genes in PDAC (e.g. *MYC*) and added coverage for them as well.

The details above have been included to the Methods section “Single-nuclei library preparation and sequencing”.

Question 3: It may be helpful to discuss more about how the sequencing depth would affect the results shown, especially for others who want to use the method in the future. A down-sampling analysis may be one of the choices.

Thank you for this great suggestion. We have done the following to address this point:

First, we included more computational pipeline metadata (e.g. scDNA-seq output mean read-depth/cell/amplicon) for each sample in **Supplementary Table 1**.

Then, we selected 2 samples (PA04-2-frozen, PR05-3) that we deemed as over-sequenced:

- (1) total number of read pairs was 278 million and 341 million, respectively.
- (2) mean read-depth/cell/amplicon was 168 and 342, respectively.

and reran the Tapestry pipeline on subsampled raw FASTQ files with intervals of 50-million reads.

Our key findings are:

- (1) with default quality control settings of the pipeline, an elbow point with respect to both the total number barcodes detected and the number of cells that passed quality control seems to appear at a mean read-depth/cell/amplicon of 100. This elbow point is more obvious with the percentage of DNA read pairs assigned to cells. A caveat is that this number could be panel-specific because of the way quality control works; the most limiting filter requires a barcode to have ample reads covering 80% “working” amplicons (pre-filtered with a total reads cutoff). Therefore, the distribution of amplicon performance as well as the size of the panel could both be factors that determine the elbow point.
- (2) On the other hand, we found out that the percentage of tumor nuclei (defined as those genotyped as mutated for *KRAS*) did not vary much with increasing read depth. This indicates that the relative clonal proportions of cells/nuclei and associated biological conclusions could be robust despite the technical parameters mentioned above. We acknowledge that this does not hold true for the purpose rare-clone discovery. Based on the subsampling experiments here, the more depth the better.

More detailed discussion and associated figures have been incorporated into the updated manuscript.

Reviewer #4:

The primary goal of this work is design of time efficient and cost effective workflow for single nuclei extraction from solid tumor tissues, which is then followed by library preparation and DNA sequencing steps yielding single-nucleus DNA sequencing data that can be used for various types of analyses (e.g, variant detection, deciphering tumor subclonal composition, studying evolutionary history and others). The authors employ this workflow to prepare 38 single-nucleus DNA libraries from 34 samples collected from 16 patients diagnosed with pancreatic ductal adenocarcinoma (PDAC). On average, 2867 nuclei per sample were sequenced using targeted 186-amplicon panel designed to cover the most frequently mutated genes in this type of cancer. Similar studies of a larger scale (both in terms of the number of patients and the number of cells sequenced per patient) were published recently (e.g., Morita et al. from 2020), but their focus is on analyzing liquid malignancies where obtaining samples is typically easier than in solid tumors. It is worth noting that the authors acknowledge and refer to these related studies, but they also present arguments for the importance of their workflow presented in this work.

Due to my background, my review will mostly focus on the data analysis part of this work. I leave the assessment of the novelty, relevance and importance of some parts of this work, such as the cells extraction, to reviewers more familiar with wet lab.

Overall, I am positive about this work and recommend revision that would address my concerns raised below, except for possibly not performing additional experiments to address suggestions related to copy number aberrations detection if such experiments would be too laborious and/or require extensive additional resources.

We appreciate your positive feedbacks.

Major Comments:

Question 1: To start with, it seems that the github link provided in the paper is broken (<https://github.com/haochenz96/mosaic>) so I could not have a look into the codes used for performing data analysis. Consequently, I am not able to make any comments regarding them. Perhaps there is an access issue or the provided link is incorrect. I also leave the possibility that something is wrong on my end. Can you please have a look into this and make sure that the code is accessible (and well documented) for the next round of review.

We apologize for the inconvenience- we made the Github repository private while we were revamping our code for an ongoing project. It is now publicly available again with documentation added.

Question 2: Page 7, the last paragraph: why do you focus only on coding variants when assessing the concordance between the sets of variants called on bWES and snDNAseq data? Or, is it that, due to the design of the panel, any variant in the regions covered by the panel is considered to be coding by default hence mention of the word "coding"?

Thank you for raising this point. While we did analyze both coding and noncoding variants we recognize that the original description of our analysis is misleading. We have now updated the text to better reflect our intent and findings.

Our intention is to see if SNVs called by bWES that are on the panel, coding or noncoding, are detected in Tapestry for validation of the pipeline. We put the

unfiltered coding SNVs' (**Figure 2a**) and filtered SNVs' (**Figure 2b**) comparison in the main text while that of all unfiltered variants in **Supplementary Figure 4A** (of revised manuscript) mostly due to space limits. We prioritize the coding SNVs also because we spotted many likely artifacts (e.g. homopolymers) enriched in the noncoding SNVs called by bWES; we also believe coding SNVs hold more functional significance. Same thing for the filtered SNVs, which were called from bWES with more confidence.

Question 3: It appears inappropriate to contrast single-cell data with “low-depth/low-coverage bulk data”; it might be more appropriate to compare single-cell data with high-quality bulk data at similar economic costs.

Thank you for the suggestion. To clarify, our bulk sequencing data is in fact high quality. We regret the wording used to imply that data was flawed when in fact many publications from our lab are based on those data. We have made the changes in text to clarify our comparison.

Question 4: [minor, but I place it here due to the comment below] Page 8, I to some extent agree that detecting a homozygous deletion present in around 30% of cells might be challenging from low quality bulk sequencing data (this certainly also depends on the size of the deletion and several other factors). But, in this particular context, is it appropriate and fair to contrast single-cell data with low-depth/low-coverage bulk data? I think that more appropriate comparison would be to compare single-cell data with high quality bulk data considering resources required to generate each. Even if the costs of obtaining each are comparable, it should be kept in mind that bulk data would usually yield much broader coverage. Shortly, while I appreciate that you conduct some preliminary experiments to demonstrate potential of detecting some (subclonal) copy number aberrations from the data generated in this work, I would recommend to rewrite this part by either completely removing any mention of bulk data or slightly toning down these claims and providing fairer comparison.

Furthermore, more importantly and related to the previous comment, while homozygous deletions represent one of the simpler-to-detect types of copy number aberrations when analyzing single-cell data, it is not clear how reliably some more challenging copy number aberrations can be detected from data generated using this pipeline (e.g., aberrations where exactly one copy of maternal and one copy of paternal allele is gained). Some analysis is shown on Pages 11, 22 and 23, but, overall, I think that the manuscript would benefit if more attention is devoted to discussing these and if some

related analyses and experiments are conducted, if possible (I understand that such analyses also depend on the extent of copy number aberrations present in the collected samples and availability of alternative means of validating them). Note that I am aware that detecting copy number aberrations is challenging, especially from targeted single-cell data, so I do not necessarily expect most of the results to be positive. However, providing more comprehensive discussion of which copy number aberrations are reliably detectable from this type of data can be very useful to the potential readers of this manuscript.

Thank you for this excellent recommendation, and we agree calling of CNVs is of great interest in single cell data.

In this paper, our only goal for CNV calling was proof-of-principle hence we implemented a naïve approach (simple normalization across cells and amplicons) to calculate ploidy which we believe showed CNV clones distinguishable by eye (e.g. homozygous deletion and LOH) despite all the noise entailed by the high-throughput PCR process.

We definitely recognize that the present approach lacks the ability to detect more challenging copy number states (e.g. a one-copy gain that you mentioned), and are actively working in collaboration with the Benjamin Raphael group at Princeton University to develop computational method to not only call such CNV clones, but also construct SNV and CNV-aware single-cell phylogenies from Tapestri data.

Specifically:

1. We have designed a larger panel (~600 amplicons) with more amplicons added for genes/genomic regions important for PDAC (*GATA6*, *MYC*, *CDKN2A*, *KRAS* etc.), because that not only enables more expansive coverage of the genome, but also enables us to leverage the read counts of more amplicons in the same genomic region to make CNV calling more robust to noise.
2. We have sequenced a large cohort of normal samples to train amplicon-specific read count models to address the heterogeneity across amplicons (i.e. different amplicons have different annealing rate, dropout rates which generate different distributions of read count data).
3. We have paired each sample's Tapestri data with 80X WGS data generated for the same samples and called genome-wide CNVs combining WGS data of multiple samples of the same patient using a state of the art tool (Holistic Allele-specific Tumor Copy-number Heterogeneity, HATCHet). We believe this will allow us to benchmark our CNV calling using Tapestri data.
4. Additionally, we are ready to use orthogonal methods (e.g. FISH, optical genome mapping) to validate the key findings we may have from above.

We believe that the abovementioned work would require too much content to illustrate for this manuscript and would go beyond the purpose of proof-of-principle; on the other hand, it wouldn't make a complete story to only include certain parts. Therefore, we plan to publish it in our next manuscript along with more in-depth biological insights or as an entirely separate technical manuscript.

We hope this is agreeable.

Question 5: Would it be possible to reconstruct and present one or a few possible trees showing clonal evolution and metastatic seeding events for the case discussed on Pages 12 and 13? Although the cohort size is very limited to study metastasis on a large scale, could you make any potentially interesting observations about the metastatic seeding in the two patients (PA03 and PA04) for which sequencing data from both, primary and metastatic site(s), were generated?

Thank you for the suggestion – we recognize that a single-cell phylogeny is perhaps the most fascinating output to generate from scDNA-seq data of multiregional samples of metastatic cancer. As detailed above, we are in the process of developing a more robust and precise CNV-calling tool and hope to combine it with SNV data to infer phylogeny with highest confidence. Additionally, we are building and testing a variant calling pipeline that will enable *de novo* SNV calling (on top of matched bulk WES/WGS) because the current one is lacking in this aspect (as mentioned in the discussion section of the main text). We are also in the process of sequencing more multiregional samples in order to get a full picture of clonal dynamics in metastasis.

Given that both the SNV and CNV pipelines are under development, we don't think a phylogeny modeled with the current data would demonstrate the full potential of the scDNA-seq technology. And that is why for the last section of analysis of the multiregional autopsy case PA04, we tried to be very conservative with our conclusion and only plotted a fish-plot showing the major clone evolution pattern.

We are aware that past papers^{5,6} have included such phylogeny analysis, but we also believe the phylogeny inference methods they implemented were not optimal. We also recognize that a tool has been developed specifically for combining SNV and CNV outputs of Tapestry data for inferring phylogeny⁷, but upon close investigation and testing we believe the tool lacks in robust SNV calling while the CNV calling needs more validation. Therefore, we are hesitant about including trees generated by that tool and are committed to developing our own methods.

Question 6:

(1) Please provide some additional clarification and motivation for the sentence "Finally, we set the threshold for the alternative (mutant) read count to 3 reads to convert low-quality heterozygous calls back to wildtype to arrive at the final cell-variant genotype matrix".

The [alternate (mutant) read count ≥ 3] filter was originally adapted from bulk-sequencing data filtering.

This also came from our observation that many false positive SNVs that are caused by misalignment/homopolymer have a low DP (<10) and a low alternate read count. Given the [DP ≥ 5] and [VAF ≥ 20] filters already applied beforehand, this alternate read filter thus only applies to cell-SNV pair with DP within [5, 10] (because an SNV with DP > 10 , VAF ≥ 20 will obviously have alternate read ≥ 3).

However, we acknowledge that this whole filtering scheme is imperfect and still outputs many false positives; for real SNVs (as validated by bulk sequencing), the DP and alternate read filters might filter out real mutant genotype in likely tumor cells. This is mainly because the default variant calling pipeline relies on GATK HaplotypeCaller with no filtering (e.g. variant quality score recalibration, VQSR) applied. We included a discussion regarding this issue in the "Discussion" section of the original manuscript.

Therefore, we are currently developing and testing a new short variant (SNV and indels) calling pipeline that relies on Mutect2 and its inherent filtering mechanisms as well as a panel of normal filtering mechanism.

The explanation has also been added to the manuscript.

(2) Also clarify what "prevalence" means in "... mutational prevalence of 0.5%" (i.e. provide a clear definition of how prevalence is calculated).

Thank you for catching this. The definition of "mutational prevalence" for an SNV is the proportion of cells having either heterozygous or homozygous genotype assigned to that SNV. We have made the clarification in the manuscript as well.

Question 7: The second paragraph of section "Single-cell per-amplicon ploidy calculation" (i.e., the paragraph starting with "Then the per-amplicon ploidy ...").

Have you made available all the information required to have this part of the analysis reproducible?

Yes. The HDF5 uploaded to EGA will have a single-cell per-amplicon read count matrix layer called “CNV”. The calculation methods would be included in the “Mosaic” package on Github and a short tutorial will be provided. We have also added these details to the “Data Availability” section.

Question 8: Lastly, I have one question related to Data availability: it is stated that all data supporting the findings of this study are in the process of uploading to EGA and will be available upon publication. Can you please comment on whether any researcher granted access to this dataset is expected to be able to reproduce all the presented results without having to make some payments to Mission Bio (e.g., paying to get premium access to access some Tapestry barcodes in order to be able to demultiplex sequencing data or having to buy an access to some data analysis software etc.)?

Thank you for pointing this out. The raw FASTQ files indeed need to be processed by the Mission Bio Tapestry pipeline into the HDF5 format which is what all the analysis in this paper was based on. The analysis from HDF5 forward can be done with the “mosaic” package which has been posted on the github page mentioned above.

We recognize that the pipeline’s details would be critical for researchers interested in method development surrounding this dataset, which we believe would be essential especially since the default pipeline has lots of parameters and modules that could use more testing and optimization. However, we did not publicize the Mission Bio pipeline in our first manuscript draft because we assumed it contains their proprietary information per industry standard.

Following this review comment, we inquired with Mission Bio. Unfortunately, they confirmed that their pipeline cannot be publicized because the barcode demultiplexing module contains the barcode sequences which is proprietary. According to them, they would only provide the pipeline details (instructions and download link) “upon requests from customers or prospects”. The full email correspondence is attached. This information has also been added to the manuscript Methods section “Single-nuclei DNA library quality control, cell-calling and variant calling”.

As disappointing as this may sound, we are actively working on optimizing and testing many non-proprietary modules of their pipeline (e.g. single-cell quality control, SNV calling etc.), and are planning to publish our work soon along with our next project mentioned above (CNV clone and single-cell phylogeny inference). With this new pipeline that we are building and customizing ourselves, we plan to isolate the module that concerns proprietary information of Mission Bio as much as possible and publish the rest for peer review.

Minor comments

1. In the doublet estimation section, please be more precise by saying that "... A contains only mutations from cell line A" (do analogous for B).

Thank you. The change has been made to the supplementary document pertaining to doublet estimation.

2. At some places, for example in the first two paragraphs on Page 8, it seems that snDNA-seq and scDNA-seq appear interchangeably. Can you please comment on this (not necessarily in the main manuscript but at least in response to this review)?

We apologize for the confusion caused by our typos. We intended to use "scDNA-seq" while describing the Tapestry technology in general while use "snDNA-seq" while describing our workflow specifically. We notice that we interchanged the two phrases at a few places inappropriately and have made the changes within text.

3. Although I find this paper to be well written, I noticed some minor typos and several examples of them are provided below.

- in the abstract, should it be "pancreatic ductal adenocarcinoma" at the place where abbreviation PDAC is first introduced?

Yes. We corrected accordingly.

- "single cell partitioning systems" -> "single-cell partitioning systems"

We corrected accordingly.

- "nuclei clumping %" -> "nuclei clumping percentage"

We corrected accordingly.

Please revisit the following:

- Page 10 "... we identified the major tumor clone carry the ..."

We corrected the grammar use.

- Page 15 "we could already see many complex clonal structures than could not be defined by driver SNV clones alone"

We corrected the grammar use.

- Page 17 "Detailed patient, sample information ..."

We corrected accordingly.

- Is comma missing after "microfluidics cartridge" (previous to the last line on Page 19)?

Absolutely. The comma has been added.

References

1. Chang, M. T. *et al.* Identifying recurrent mutations in cancer reveals widespread lineage diversity and mutational specificity. *Nat. Biotechnol.* **34**, 155–163 (2016).
2. Bailey, M. H. *et al.* Comprehensive Characterization of Cancer Driver Genes and Mutations. *Cell* **173**, 371-385.e18 (2018).
3. Reiter, J. G. *et al.* Reconstructing metastatic seeding patterns of human cancers. *Nat. Commun.* **8**, 1–10 (2017).
4. Reiter, J. G. *et al.* An analysis of genetic heterogeneity in untreated cancers. *Nat. Rev. Cancer* **19**, 639–650 (2019).
5. Miles, L. A. *et al.* Single-cell mutation analysis of clonal evolution in myeloid malignancies. *Nature* **587**, 477–482 (2020).
6. Leighton, J., Hu, M., Sei, E., Meric-Bernstam, F. & Navin, N. E. Reconstructing mutational lineages in breast cancer by multi-patient-targeted single cell DNA sequencing. *bioRxiv* 2021.11.16.468877 (2021)
7. Sollier, E., Kuipers, J., Takahashi, K., Beerenwinkel, N. & Jahn, K. Joint copy number and mutation phylogeny reconstruction from single-cell amplicon sequencing data. *bioRxiv* 2022.01.06.475205 (2022)

REVIEWER COMMENTS

Reviewer #2 (Remarks to the Author):

Thank you for addressing my previous concerns! I don't have further comments.

Reviewer #4 (Remarks to the Author):

Thank you for revising the original manuscript and addressing most of my concerns. After reviewing your responses, I only have the following two questions and one request:

1. Related to Question 6, do you have estimates of false positive and false negative rates in SNV calling (if it helps, assume for simplicity that HOM and HET are treated as a single (i.e., "present") category)?
2. Unfortunately I am not very satisfied with the answer to Question 5 and I do not find the provided response to be convincing. While in my first review I have suggested some analysis of metastatic seeding, I believe that, in addition to some potential biological insights, phylogenetic analysis can also give some insights about the generated data. For example, assume that a phylogenetic trees are obtained for the entire set of the sequenced patients. Tree reconstruction typically requires error-corrections (flipping entries of the input genotype matrix) to correct for false positives and false negatives. If, for example, at some amplicon no copy number aberrations are observed, but the numbers of corrections for false negatives (0->1 flips) are considerably higher compared to average 0->1 flip counts, then this might be an indication of elevated allelic dropout rates at the amplicon, suggesting potential biases in PCR amplification, poor primer binding etc. Note that phylogenetic principles have also been directly leveraged in the design of some variant calling methods (see <https://doi.org/10.1038/s41467-018-07627-7> or <https://doi.org/10.1093/bioinformatics/btac254>). Could you please revisit this? Please note that I am not asking for a sophisticated phylogenetic analysis and devoting half of the manuscript to that, but running a tool like, e.g., SCITE (<https://doi.org/10.1186/s13059-016-0936-x>), and providing some biological interpretation of the results, as well as discussion of the flip patterns could benefit and strengthen this work.
3. Regarding the Question 8, I very much appreciate the efforts that you made to make all data available to the broader research community. Considering some restrictions that would still persist, can you please at least make sure that genotype matrices, which contain information about presence or absence (or missing) of variants in cells, are made available for all sequenced patients/samples, as well as the exact read counts information for each of the four nucleotides at each variant site?

MINOR

I also have some minor comments.

(i)

While Github repository is currently available and is in a decent shape, I noticed that some

parts of the documentation need some revision and corrections. I will provide some examples.

1. At https://github.com/haochenz96/mosaic/blob/simplified/tutorials/mosaic-CNV_analysis.ipynb

there is the following sentence:

"Changes were changed to optimize tumor sample analysis used for Zhang et al. (2022)."

In addition to "Changes were changed", I do not see which paper exactly Zhang et al. (2022) points to. I understand that this is a simple tutorial notebook so it is not required to be formal to an extent that a list of citations is attached at the end of the notebook, but I recommend at least adding a doi or some other link right after the reference (similarly as was done on the main README page).

The following three comments are related to the second notebook, which can be found at at

https://github.com/haochenz96/mosaic/blob/simplified/tutorials/mosaic_clone_analysis.ipynb

2. The function `sample.dna.genotype()` is introduced, its parameters are then explained and after that it is not used at all in the notebook. Perhaps you were referring to `sample_obj.dna.genotype_variants` ?

3. I was confused by the following parameter and its explanation

"min_dp [default: None]: minimum depth to assign an SNV in a cell as missing (MISSING, numbered 3)". Isn't it more intuitive that min_dp is cutoff for assigning a variant as non-MISSING? For example, if min_dp=5 then as soon as coverage is at or above this minimum threshold of 5 reads we assume that there is sufficient information to call the variant (i.e., as 0, 1 or 2).

Alternatively, one can use max_dp instead of min_dp to define maximum depth until/under which a MISSING status is assigned.

I also checked the source code at

<https://github.com/haochenz96/mosaic/blob/simplified/src/mosaic/dna.py>

and the implementation of the function `genotype_variants`:

```
def genotype_variants(self, het_vaf=20, hom_vaf=80, min_dp=None, min_alt_read = None, min_gq = 0)
```

where the following description (which I think matches my above suggestion) is provided
"min_dp : int [0, inf] minimum depth for a variant to be considered covered in one cell"

4. In

"- 1. load and genotype SNV matrix

- plot single-cell mutational-prevalence histogram

- plot single-cell heatmap for SNV data"

why is the number "1." placed in front of "load and genotype SNV matrix"?

There are some other examples, but let me stop here. In summary, please just carefully revisit the repository once again. I leave it at your discretion to address this and I do not expect detailed explanation of what was done in Response to Reviewers.

(ii)

I noticed that in Response to Reviewers you attributed me the following sentence, which I have not written in my review.

"Question 3: It appears inappropriate to contrast single-cell data with "low-depth/low-coverage bulk data"; it might be more appropriate to compare single-cell data with high-quality bulk data at similar economic costs."

However, taking into account the content of the above sentence and comparing it to what I had asked for, it is quite clear that this mistake was not made intentionally. I do not expect anything to be done regarding this.

Salem Malikic
Cancer Data Science Laboratory, NCI

Manuscript NCOMMS-22-08919A

Responses to referees

Haochen Zhang^{1,2}, Christine Iacobuzio-Donahue^{2,3,4*}

¹ Gerstner Sloan Kettering Graduate School of Biomedical Sciences, Memorial Sloan Kettering Cancer Center, New York, NY, USA.

² Human Oncology and Pathogenesis Program, Memorial Sloan Kettering Cancer Center, New York, NY, USA.

³ David M. Rubenstein Center for Pancreatic Cancer Research, Memorial Sloan Kettering Cancer Center, New York, NY, USA.

⁴ Department of Pathology and Laboratory Medicine, Memorial Sloan Kettering Cancer Center, New York, NY, USA.

*Correspondence: iacobuzc@mskcc.org

We thank the editor and reviewers for their helpful feedbacks. We highlighted/commented substantial changes made to the manuscript per reviewers' comments. Below we provide a point-by-point response (blue text) to each reviewer's comments (black text). All references to sections and figures refer to the revised manuscript, unless specified otherwise.

Reviewer #2:

Thank you for addressing my previous concerns! I don't have further comments.

We appreciate your positive feedback.

Reviewer #4:

Thank you for revising the original manuscript and addressing most of my concerns.

We appreciate your positive feedback.

After reviewing your responses, I only have the following two questions and one request:

1. Related to Question 6, do you have estimates of false positive and false negative rates in SNV calling (if it helps, assume for simplicity that HOM and HET are treated as a single (i.e., "present") category)?

Thank you for the request. We agree that these parameters would indeed be very important for researchers interested in modeling the error profile for this kind of dataset.

Unfortunately, we are unable to provide a formal calculation of false positive or false negative rates in SNV calling as there is no dataset for us to benchmark on. We considered creating a synthetic dataset and benchmarking SNV calling on that but ultimately decided it is not within the scope of this paper. However, we believe the comparison with bulk-sequencing results shown in **Figure 2** can to some extent show the false negative rate of the current SNV calling method, which we believe to be low.

Nonetheless, because we agree this is an important point we took an alternative approach by estimating the allelic dropout (ADO) rate. Ideally this should be done on normal samples which are mostly unaffected by genome-scale copy number variations (CNVs), but unfortunately, we did not include such samples in this proof-of-principle cohort. Therefore, we picked 19 early-stage PDAC resection samples as genome-scale CNVs are less prevalent than late-stage PDAC autopsy samples.

We identified germline heterozygous single nucleotide polymorphisms (SNPs) by first comparing snDNA-seq data with bulk whole-exome sequencing (WES) data and filtering for SNVs with VAF>0.2 in bulk normal sample. This was mainly to exclude technical artifacts rather than filtering for heterozygosity. We then filtered the SNV list with respect to snDNA-seq data: an SNV is only considered if its single-nucleus mean VAF (DP=0 not included in mean calculation) is between 0.2 and 0.8 across all nuclei. This was to exclude SNVs that are germline homozygous or are associated with somatic mosaicism. Although this might exclude real germline heterozygous SNPs that have high ADO rate, this filter excluded only a small proportion of SNVs from our observation and should not bias the ADO calculation much.

Finally, we calculated the ADO rate for a germline heterozygous SNP as:
(number of nuclei with strictly 0 read of either allele) / (number of total nuclei)

Based on this approach we calculated the mean ADO rate for all germline heterozygous SNPs for all 19 samples to be **0.196**. We have now included these results (main text **page 7**), additional methods (main text **page 24**) and metadata (**Supplementary Table 3**) in our updated manuscript.

2. Unfortunately I am not very satisfied with the answer to Question 5 and I do not find the provided response to be convincing. While in my first review I have suggested some analysis of metastatic seeding, I believe that, in addition to some potential biological insights, phylogenetic analysis can also give some insights about the generated data. For example, assume that a phylogenetic trees are obtained for the entire set of the sequenced patients. Tree reconstruction typically requires error-corrections (flipping entries of the input genotype matrix) to correct for false positives and false negatives. If, for example, at some amplicon no copy number aberrations are observed, but the numbers of corrections for false negatives (0->1 flips) are considerably higher compared to average 0->1 flip counts, then this might be an indication of elevated allelic dropout rates at the amplicon, suggesting potential biases in PCR amplification, poor primer binding etc. Note that phylogenetic principles have also been directly leveraged in the design of some variant calling methods (see <https://doi.org/10.1038/s41467-018-07627-7> or <https://doi.org/10.1093/bioinformatics/btac254>). Could you please revisit this? Please note that I am not asking for a sophisticated phylogenetic analysis and devoting half of the manuscript to that, but running a tool like, e.g., SCITE (<https://doi.org/10.1186/s13059-016-0936-x>), and providing some biological interpretation of the results, as well as discussion of the flip patterns could benefit and strengthen this work.

Thank you for raising this point which we have revisited based on agreement that running these tools would be helpful. A summary of our results running a set of phylogeny-building algorithms on case PA04 is as follows:

SCIPHI

We had actually attempted SCIPHI¹ before submitting the original manuscript. However, we concluded that the tool was built for first-generation single-cell DNA-sequencing datasets which consist of low cell-throughputs (<100) and thus made it inapplicable to our dataset that is an order of magnitude greater in cell number (>1000 cells per sample).

SCITE

We then attempted SCITE² per your suggestion, which seemed to work on our dataset's cell throughput, to the 5-sample (including 2 technical replicates) concatenated SNV genotype matrix of case PA04. To avoid making any assumption on *de novo* SNVs, we only included 3 somatic mutated loci validated by bulk sequencing on the same cohort (1x *KRAS*, 2x *SMAD4*, as used in our original paper). Given that SCITE does not allow a mutation to be lost, we decided not to include germline SNPs as ongoing work indicates frequent LOH events affecting these loci.

As expected, with the 3 somatic SNVs, we could only infer a linear phylogeny (**Figure R1**) similar to what our original raw heatmaps (**Figure 6 in the original manuscript**) suggested. Unfortunately it also did not provide additional technical insight with regards to, for example, allelic dropout, because there were clearly CNVs in all 3 loci as shown by our original raw heatmaps.

Figure R1: SNV clone tree inferred by SCITE of all single nuclei of case PA04

COMPASS

We next turned to COMPASS³ which is SNV-and-CNV-aware and is specifically built for the Tapestry dataset. Additionally, this tool models raw reference/alternate allele read counts for the SNV part which is highly desirable given the significant ADO rate.

For the SNV input, again to avoid making assumptions on *de novo* SNVs, we used only the 3 somatic SNVs as used for SCITE above, as well as one more somatic SNV of PIK3CA which was identified by bulk sequencing on a different metastatic site of this patient (not in this snDNA-seq study cohort) and had some signal in the snDNA-seq data. In addition, we included 21 germline SNPs validated by bulk sequencing on the matched normal sample hoping to leverage them for inferring CNV events.

For the CNV input, we thought a simple summation of read counts across multiple amplicons targeting the same gene, as used in the original paper, was not addressing inter-amplicon heterogeneity, which could be either technical or biological (focal events). Therefore, we used single-amplicon-level read counts. To avoid too much complexity, we only used the 23 amplicons targeting the 25 SNV/SNPs above, and we only used one of the 5 samples of PA04 (PA04-2-fresh) which seemed to have the most clonal heterogeneity among all 5 samples as our test case.

Because it was impossible to assess convergence of the MCMC-based method, we let COMPASS run with 8 chains each of length 100,000 which is significantly longer than used in the original paper. Despite our attempts to include a small subset of our data, it still produced a complex tree (**Figure R2**) with obvious errors with respect to the 3 somatic SNVs:

- (1) the obviously clonal KRAS p.G12V event is placed close to the end of one lineage (left side of the **Figure R2**);
- (2) the obviously subclonal SMAD4 p.D52Rfs2 event is placed at the root of the tree.
- (3) the obviously subclonal homozygous deletion of AMPL84306 (where the 2 somatic SMAD4 SNVs are located) is resolved as multiple independent CNV events, which include copy neutral LOH (CNLOH), 1-copy-gain, 1-copy-loss.

These obvious errors, plus the lack of interpretability of the program (this software was not developed with much supportive information on how to interpret the outputs) made us hesitant about moving forward with this tool.

Figure R2: Clone tree output by COMPASS on sample PA04-2-fresh

Single-Cell Genotyper (SCG)

We then attempted SCG⁴, which seemed to be scalable to our dataset's cell throughput, and the clustering algorithm allows for:

- (1) variable number of data states, which enables differentiating among heterozygous, homozygous, and missing SNV genotypes.
- (2) variable layers of data, which enables inclusion of both SNV and CNV events. while not making any assumption of the evolutionary relationships among clusters.

Below is a brief summary of our methods. Complete details can now be found in the **Supplementary Methods** section of our updated manuscript.

For the SNV input matrix, we defined 4 states: 0 as wildtype (WT), 1 as heterozygous (HET), 2 as homozygous (HOM), 3 as missing (MISS); In addition to the 4 somatic SNVs mentioned above, we also included germline SNPs validated by bulk on the matched normal sample of this patient, since their SNV genotype could represent CNV events. The genotype of SNV in each single cell was assigned based on hard thresholds:

For an SNV to be considered HET in a cell, it needs to satisfy:

- (1) alternative allele read count ≥ 0
- (2) variant allele frequency (VAF) > 0.2

For an SNV to be considered HOM in a cell, in addition to the two requirements for HET above, it needs to satisfy:

- (3) VAF > 0.8

For an SNV to be considered MISS in a cell, it needs to satisfy:

- (1) total read depth = 0

The thresholds were set to be slightly more lenient than used in the original manuscript (except for MISS) because we hoped to push the error correction to SCG.

For the CNV input matrix, to avoid the complexity of accurate copy number calling, we only included the homozygous deletion status (0 as ploidy >0 , 1 as ploidy=0). This was based on our assumption that a read count of 0 very likely (99% as defined in the hyperparameters) represents a real homozygous deletion.

Ploidy status was determined by a hard threshold, too: for an amplicon to be ploidy=0, it needs to strictly satisfy that the forward read count equals 0.

To focus on studying *SMAD4*'s CNV evolution, we only included *SMAD4*'s 8 amplicons. Based on our raw data we hypothesized that the 8 amplicons underwent a likely asynchronous, step-wise process (this was also stated in the original manuscript) to the final state of 8/8 homozygous deletion, assuming that a deleted fragment of DNA cannot be regained.

We started by concatenating 8061 cells from all 5 samples (including 2 technical replicates) of patient PA04 to do the clustering. We set the number of clusters to 40, with the belief that clonal selection would be almost complete in such a late-stage PDAC case and the number of observable/computationally solvable clone with respect to the input loci/regions should be

less than 40. Other parameters, including the emission hyperparameters, have been detailed in the **Supplementary Methods** of the updated manuscript.

After running SCG with 5000 random restarts, we recognize that the model is highly subject to local maxima. See below (**Figure R3**) for the comparison of the evidence lower bound (ELBO) plots of 2000 and 5000 random restarts. The max ELBO improved very minimally and did not seem to be converging. Therefore, we did not move upwards from 5000 restarts.

Figure R3: Evidence lower bound (ELBO) plots of SCG: 2000 restarts vs 5000 restarts

With the clustering results, we defined a “genomic evolution cost” matrix for every SNV and CNV state transitions between each pair of unique clusters (**Figure R4**) and inferred the minimum spanning arborescence using a modified Edmond’s algorithm.

Figure R4: CNV and SNV state transition matrices used for inferring the clone tree from SCG clusters.

Encouragingly, the 5-sample combined run's solution gave only 7 unique clones (**Figure R5**), which constituted a largely linear clone tree (**Figure R6**). The evolution process aligned with what was described in our original manuscript but with some additional details:

- First, the 2 major somatic SNVs (*KRAS* p.G12V, *SMAD4* p.A39T) occurred.
- Second, large-scale CNVs occurred which affected *TP53*, *KRAS*, *AXIN* and several genes on chromosome 3.
- Ongoing CNV likely created clonal heterogeneity:
 - (1) one minor (as assessed by clonal prevalence) branch went into focal deletion of only one of *SMAD4*'s amplicons region and a part of *RREB1*.
 - (2) the major branch went into deletion of more of *SMAD4*'s genomic region (homdel of 3 amplicons, LOH of the region carrying the somatic SNV). This branch further deleted more *SMAD4*'s genomic region which resulted in the most dominant clone 0 in the entire 5-sample cohort, characterized by homdel of 7/8 *SMAD4* amplicons.

This validated our hypothesis that *SMAD4*'s 8 amplicons underwent step-wise focal deletions. Interestingly, clone5, which diverged from main lineage and was characterized by focal *RREB1* homdel, might reveal convergent evolution towards inactivation of the anti-tumor mechanism of tumor cell-intrinsic TGF- β signaling⁵⁻⁸.

Figure R5: profiles of clusters identified by SCG on all 5 samples of PA04.

Left panel: number of cells comprising each cluster across the entire cohort and their corresponding cluster number and color. The cluster numbers were assigned based on cluster size with the biggest cluster assigned as 0; the cluster colors were assigned to maximize heterogeneity. Middle panel: 8 *SMAD4* amplicons' homozygous deletion (homdel) status. Right panel: 25 SNVs' genotype. The 4 somatic SNVs identified by bulk sequencing in this patient's samples are colored red.

Figure R6: clone tree inferred for clusters identified by across all 5 samples of PA04.

Cluster-defining genomic events are labeled on each cluster. The proportion of single nuclei that each cluster comprises across the cohort is labeled as the percentage next to the cluster ID. The border color of each cluster matches that in **Figure R5**.

A few important clones that we identified from the raw data, such as the one with the second somatic *SMAD4* SNV (chr18:48573563:G/GA) and the one with homdel of the *SMAD4* amplicon carrying the two somatic *SMAD4* SNV's, were unfortunately not identified by SCG likely due to their low prevalence within the entire cohort. Therefore, to delve into those clones' evolution history, we selected one of the 5 samples – PA04-2-fresh (2298 cells) - which seemed to carry those minor clones, and ran SCG again with all other parameters fixed.

We were able to identify 13 unique clones with this run (**Figure R7**). As we tried tree construction, we noticed that the 3 smallest clones were likely technical artifacts due to genotyping error and were confounding the tree structure. We therefore removed them and arrived at a cleaner tree structure (**Figure R8**). After all the clonal events such as the 2 somatic SNVs, *TP53* LOH etc., we could see 2 diverging lineages where the first, dominated by clones #0 and #2, went in the direction of homdel of more *SMAD4* genomic regions and ended up at clone #3 where all 8 amplicons' genomic region was deleted; the second, dominated by clones #1 and #4, went in the direction of focal homdel of less of *SMAD4* but focal deletion of *RREB1*. Intriguingly, the first lineage also ended up with focal deletion of *RREB1* (clones #2, #8), which may be caused again by inaccurate genotyping of the *RREB1* germline SNP, or represented a real biological phenomenon of convergent focal deletion of the same *RREB1* genomic region.

Figure R7: profiles of clusters identified by SCG on PA04-2-fresh

The panels and cluster numbers were constructed in the same way as described for **Figure R5**. The cluster colors were assigned also based on cluster prevalence, with bigger clusters being more red and smaller clusters more blue.

Figure R8: clone tree inferred for clusters identified in PA04-2-fresh, excluding smaller confounding clones.

The tree was not as well annotated as **Figure R6** to avoid the complexity added by increased number of clusters.

Conclusion and next-step plan

Ultimately, we believe the pan-cohort SCG run result was the cleanest, validated and extended our observation of the evolutionary process from the raw data, and added in the important insight that two lineages of tumor cells might have undergone convergent evolution to TGF- β pathway inactivation. Therefore, we added this result as a new **Figure 6** in the manuscript, while moving the original **Figure 6** to **Supplementary Figure 9**. We also updated our **Results, Methods, Supplementary Methods** accordingly.

We decide not to include the single-sample run results (**Figure R7, R8**) because it carries a bit too much ambiguity, which we are aiming to address in our next publication. We included it here for transparency.

Thank you again for this comment which has made our manuscript better.

3. Regarding the Question 8, I very much appreciate the efforts that you made to make all data available to the broader research community. Considering some restrictions that would still persist, can you please at least make sure that genotype matrices, which contain information about presence or absence (or missing) of variants in cells, are made available for all sequenced patients/samples, as well as the exact read counts information for each of the four nucleotides at each variant site?

Thanks for this great suggestion. To address this, we will upload pre-filtered BAM files, which are aligned read sequences assigned to read groups corresponding to the associated single-droplet barcode sequences identified. This step is before the “cell-finding” step, which QC’s each single-droplet barcode’s associated reads based on a total read cutoff and a pan-amplicon performance cutoff. After the QC step, the BAM file can be split into individual read group, which corresponds to each single cell, for variant calling with GATK HaplotypeCaller. We believe this should provide the level of data you are interested in.

We would also like to notify you of an update related to the Tapestri pipeline. A few weeks ago when we were reviewing one of the original papers on the Tapestri single-cell DNA sequencing technology, which happens to be published in the *Nature Communications* journal as well (<https://www.nature.com/articles/s41467-021-21810-3>), we noticed that the paper’s first author Dr. Demaree developed and publicized (<https://github.com/AbateLab/DAb-seq>) what we believe to be an earlier version of Mission Bio’s default pipeline for processing Tapestri single-cell DNA sequencing data. Although we have not formally tested to confirm that the two version perform identically, upon inspection of the code, we think the core algorithms (alignment, barcode extraction, “cell-finding” [quality control], SNV calling) are essentially the same, and the publicized Github repository contains the single-cell barcodes which Mission Bio claimed to be proprietary. We believe Dr. Demaree’s Github repository would provide a comprehensive guide to researchers interested in replicating our results and understanding the upstream data processing workflow of Tapestri data, and will add a link to his site in our methods section. We hope you find this acceptable.

MINOR:

(i) While Github repository is currently available and is in a decent shape, I noticed that some parts of the documentation need some revision and corrections. I will provide some examples.

Thank you for your careful review of my tutorials. I sincerely apologize for the inconvenience. I am on the way to fix every issue you mentioned and adding more details to enhance user experience. Please feel free to open Github issues if you have more suggestions. However, since the repo is maintained entirely by myself (HZ) and I am quite a novice in software development, I cannot guarantee immediate responses. I hope you could understand.

1. At https://github.com/haochenz96/mosaic/blob/simplified/tutorials/mosaic-CNV_analysis.ipynb there is the following sentence:

"Changes were changed to optimize tumor sample analysis used for Zhang et al. (2022)."

In addition to "Changes were changed", I do not see which paper exactly Zhang et al. (2022) points to. I understand that this is a simple tutorial notebook so it is not required to be formal to an extent that a list of citations is attached at the end of the notebook, but I recommend at least adding a doi or some other link right after the reference (similarly as was done on the main README page).

Thank you. A link has been added.

The following three comments are related to the second notebook, which can be found at https://github.com/haochenz96/mosaic/blob/simplified/tutorials/mosaic_clone_analysis.ipynb

2. The function `sample.dna.genotype()` is introduced, its parameters are then explained and after that it is not used at all in the notebook. Perhaps you were referring to `sample_obj.dna.genotype_variants` ?

Thank you. The typo has been fixed.

3. I was confused by the following parameter and its explanation

"min_dp [default: None]: minimum depth to assign an SNV in a cell as missing (MISSING, numbered 3)". Isn't it more intuitive that min_dp is cutoff for assigning a variant as non-MISSING? For example, if min_dp=5 then as soon as coverage is at or above this minimum threshold of 5 reads we assume that there is sufficient information to call the variant (i.e., as 0, 1 or 2).

Alternatively, one can use max_dp instead of min_dp to define maximum depth until/under which a MISSING status is assigned.

I also checked the source code at

<https://github.com/haochenz96/mosaic/blob/simplified/src/mosaic/dna.py>

and the implementation of the function `genotype_variants`:

```
def genotype_variants(self, het_vaf=20, hom_vaf=80, min_dp=None, min_alt_read = None, min_gq = 0)
```

where the following description (which I think matches my above suggestion) is provided

"min_dp : int [0, inf] minimum depth for a variant to be considered covered in one cell"

Thank you for careful consideration. I recognize that my logic was wrong and has fixed the description in the notebook. You are right in that min_dp refers to the minimum depth to assign an SNV in a cell as *non-missing*.

4. In

"- 1. load and genotype SNV matrix

- plot single-cell mutational-prevalence histogram

- plot single-cell heatmap for SNV data"

why is the number "1." placed in front of "load and genotype SNV matrix"?

Thank you for pointing this out. It has been fixed.

There are some other examples, but let me stop here. In summary, please just carefully revisit the repository once again. I leave it at your discretion to address this and I do not expect detailed explanation of what was done in Response to Reviewers.

(ii)

I noticed that in Response to Reviewers you attributed me the following sentence, which I have not written in my review.

"Question 3: It appears inappropriate to contrast single-cell data with "low-depth/low-coverage bulk data"; it might be more appropriate to compare single-cell data with high-quality bulk data at similar economic costs."

However, taking into account the content of the above sentence and comparing it to what I had asked for, it is quite clear that this mistake was not made intentionally. I do not expect anything to be done regarding this.

We apologize for this mistake- the paragraph came from a draft version of our letter where we paraphrased your original question (immediately following this paragraph) for simplicity, and we forgot to delete it.

References:

1. Singer, J., Kuipers, J., Jahn, K. & Beerenwinkel, N. Single-cell mutation identification via phylogenetic inference. *Nat. Commun.* **9**, 1–8 (2018).
2. Kuipers, J., Jahn, K., Raphael, B. J. & Beerenwinkel, N. Single-cell sequencing data reveal widespread recurrence and loss of mutational hits in the life histories of tumors. *Genome Res.* **27**, 1885–1894 (2017).
3. Sollier, E., Kuipers, J., Takahashi, K., Beerenwinkel, N. & Jahn, K. Joint copy number and mutation phylogeny reconstruction from single-cell amplicon sequencing data. *bioRxiv* 2022.01.06.475205 (2022) doi:10.1101/2022.01.06.475205.
4. Roth, A. *et al.* Clonal genotype and population structure inference from single-cell tumor sequencing. *Nat. Methods* 2016 137 **13**, 573–576 (2016).
5. Iacobuzio-Donahue, C. A. *et al.* DPC4 gene status of the primary carcinoma correlates with patterns of failure in patients with pancreatic cancer. *J. Clin. Oncol.* **27**, 1806–1813 (2009).
6. Herman, J. M. *et al.* Smad4 Loss Correlates with Higher Rates of Local and Distant Failure in Pancreatic Adenocarcinoma Patients Receiving Adjuvant Chemoradiation. *Pancreas* **47**, 208–212 (2018).
7. Massagué, J. TGF β signalling in context. *Nature Reviews Molecular Cell Biology* vol. 13 616–630 (2012).
8. Su, J. *et al.* TGF- β orchestrates fibrogenic and developmental EMTs via the RAS effector RREB1. *Nature* **577**, 566–571 (2020).

REVIEWERS' COMMENTS

Reviewer #4 (Remarks to the Author):

I thank the Authors for addressing (most of) my comments.

I noticed that some of them were not addressed. For example, the following (minor) comment:

" At https://github.com/haochenz96/mosaic/blob/simplified/tutorials/mosaic-CNV_analysis.ipynb there is the following sentence: "Changes were changed to optimize tumor sample analysis used for Zhang et al. (2022)." In addition to "Changes were changed", I do not see which paper exactly Zhang et al. (2022) points to. I understand that this is a simple tutorial notebook so it is not required to be formal to an extent that a list of citations is attached at the end of the notebook, but I recommend at least adding a doi or some other link right after the reference (similarly as was done on the main README page)."

[Authors response] Thank you. A link has been added.

I see no link. In addition, the comment is not only about the missing link but "Changes were changed" sounds ambiguous (and it is still there). Perhaps you forgot to push/upload your code updates to the repository.

Anyway, since these are minor points, I leave it at your discretion to make sure that all claims from Response to Reviewers hold true.

Thank you for addressing my comments and good luck with the publication. I have no further comments.

Salem Malikic
National Cancer Institute

Manuscript NCOMMS-22-08919B

Responses to referees

Haochen Zhang^{1,2}, Christine Iacobuzio-Donahue^{2,3,4*}

¹ Gerstner Sloan Kettering Graduate School of Biomedical Sciences, Memorial Sloan Kettering Cancer Center, New York, NY, USA.

² Human Oncology and Pathogenesis Program, Memorial Sloan Kettering Cancer Center, New York, NY, USA.

³ David M. Rubenstein Center for Pancreatic Cancer Research, Memorial Sloan Kettering Cancer Center, New York, NY, USA.

⁴ Department of Pathology and Laboratory Medicine, Memorial Sloan Kettering Cancer Center, New York, NY, USA.

*Correspondence: iacobuzc@mskcc.org

We thank the editor and reviewers for their helpful feedbacks. We highlighted/commented substantial changes made to the manuscript per reviewers' comments. Below we provide a point-by-point response (blue text) to each reviewer's comments (black text). All references to sections and figures refer to the revised manuscript.

Reviewer #4

I thank the Authors for addressing (most of) my comments.

Our pleasure. Thank you for helping us improve our work.

I noticed that some of them were not addressed. For example, the following (minor) comment: " At https://github.com/haochenz96/mosaic/blob/simplified/tutorials/mosaic-CNV_analysis.ipynb there is the following sentence: "Changes were changed to optimize tumor sample analysis used for Zhang et al. (2022)." In addition to "Changes were changed", I do not see which paper exactly Zhang et al. (2022) points to. I understand that this is a simple tutorial notebook so it is not required to be formal to an extent that a list of citations is attached at the end of the notebook, but I recommend at least adding a doi or some other link right after the reference (similarly as was done on the main README page)."

[Authors response] Thank you. A link has been added.

I see no link. In addition, the comment is not only about the missing link but "Changes were changed" sounds ambiguous (and it is still there).

Perhaps you forgot to push/upload your code updates to the repository.

We sincerely apologize for the mistake. We have again updated our tutorial notebook. Please feel free to email us directly or open Github issues should you have any more suggestions.

Anyway, since these are minor points, I leave it at your discretion to make sure that all claims from Response to Reviewers hold true.

Thank you for addressing my comments and good luck with the publication. I have no further comments.

Again, it is our great pleasure to receive your helpful comments. They have improved our manuscript significantly.